# ROUTING-DECONSTRUCTED LORA IN FEDERATED FINE-TUNING

## ABSTRACT

The integration of Large Language Models (LLMs) with Federated Learning (FL) offers a promising approach to privacy-preserving Parameter-Efficient Fine-Tuning (PEFT). However, resource and data heterogeneity in FL cause differences in local knowledge distribution across clients. As a representative PEFT approach, LoRA still faces three key challenges in such settings: ***aggregation noise***, ***knowledge contamination***, and ***aggregation distortion***. To address these issues, we propose Routing-Deconstructed LoRA (RD-LoRA). Building on an alternating freezing strategy to mitigate ***aggregation noise*** and concurrently reduces communication cost, RD-LoRA further introduces two novel components. For ***knowledge contamination***, we design a Server-Client Routing Deconstructor (SCRD) that separates shared semantics from local biases, retaining fine-grained knowledge with semantic consistency. To address ***aggregation distortion***, we propose a Poly-Consensus Aggregation (PCA) mechanism that uses adaptive weighted averaging to align global LoRA parameters with heterogeneous client distributions, thus correcting the global update direction. Extensive experiments demonstrate that RD-LoRA is effective and robust in both homogeneous and heterogeneous settings.

## 1 INTRODUCTION

Foundation Models (Devlin et al., 2019; Dosovitskiy et al., 2020; Lyu et al., 2023; Zhou et al., 2024) contain hundreds of millions or billions of parameters and, once fine-tuned, provide strong initial weights for diverse downstream tasks (Wang et al., 2018; Wu et al., 2023), but their scale makes fine-tuning costly (Liu et al., 2021; Ye et al., 2024). PEFT addresses this by updating a small subset of parameters or adding lightweight modules, such as prompt tuning, LoRA, and BitFit (Lester et al., 2021; Hu et al., 2022; Zaken et al., 2021), which greatly reduces the computation. Owing to this efficiency, PEFT has been integrated with FL to enable privacy-preserving collaborative updates by avoiding the exchange of raw data, thereby further reducing communication and computation costs (Babakniya et al., 2023; Sun et al., 2024; Zhang et al., 2024a). Among PEFT methods, LoRA exhibits strong performance in medium-scale and domain-shifted tasks (Zhang et al., 2023b), inspiring federated variants such as FedIT for instruction tuning (Zhang et al., 2024a) and SLoRA, which incorporates an initialization strategy that accelerates convergence (Babakniya et al., 2023).

However, directly applying LoRA in real FL introduces challenges, most notably ***aggregation noise***. In standard federated LLM fine-tuning (Ye et al., 2024), modules A and B are averaged separately across clients, causing the aggregated LoRA to deviate from the ideal global model and inject noise into updates. As shown in Figure 1, two main solutions are used: (1) Merging A and B (Wang et al., 2024; Bai et al., 2024), where the matrices are multiplied to form B and A and then aggregated; (2) Alternating training of A and B (Koo et al., 2024; Salami et al., 2024), where only the one module (*i.e.*, A or B) is frozen per round and the other is aggregated. Compared with "Merging", "Alternating" nearly halves the communication, since only A or B is transmitted each round. The "Alternating" method (Koo et al., 2024) reduces communication by selecting weights from a globally initialized LoRA with uniform rank. This homogeneity assumption conflicts with the resource heterogeneity of FL (Imteaj et al., 2021), where clients typically adopt different initial ranks.

In real-world FL, clients often hold highly diverse non-IID data due to variations in behavior, domains, and collection conditions (Zhang et al., 2024c; Vahidian et al., 2023; Wang et al., 2022a; Yao et al., 2021). System-level resource heterogeneity further amplifies these discrepancies, driving local models towards their own optima (Imteaj et al., 2021). Traditional averaging methods such as FedAvg (Vahidian et al., 2023) assume the update directions have been aligned, but under strong

Figure 1: Comparison of different federated fine-tuning LoRAs. Compared to methods (a), (b), and (c), our RD-LoRA performs better in addressing aggregation noise, communication costs, knowledge contamination, and aggregation distortion.

heterogeneity, this assumption breaks down. However, simple averaging then leads to **aggregation distortion**, where global parameters deviate from the clients' optima and dilute local information, reducing the adaptability to various distributions (Jimenez-Gutierrez et al., 2025; Wang et al., 2023). During client-side training with heterogeneous data, fusing global parameters into client-specific knowledge spaces often causes **knowledge contamination**, as local biases mix with shared representations and hinder learning. The key challenge is to maintain global consistency while retaining and exploiting the fine-grained knowledge of each client.

To simultaneously mitigate **aggregation noise**, **knowledge contamination**, and **aggregation distortion** in heterogeneous settings, as shown in Figure 1, we propose a federated fine-tuning framework called Routing-Deconstructed Low-Rank Adaptation (RD-LoRA). Building on the alternating training of modules A and B, an strategy that primarily mitigates *aggregation noise* while also reducing communication cost, RD-LoRA further introduces two central strategies. Firstly, we design a Server-Client Routing Deconstructor (SCRD) on the client side to separate global and local knowledge. This structural decoupling effectively mitigates *knowledge contamination* in federated learning, where local information tends to be overly diluted or conflated with global information during the training process on client devices. Secondly, to alleviate *aggregation distortion*, we develop a Poly-Consensus Aggregation (PCA) mechanism during the server-side update phase. This mechanism combining adaptive consensus weights and historical regularization to stabilize server-side aggregation and keep global LoRA parameters aligned with clients' optimization trajectories. The main contributions of this study are summarized as follows:

- **Server-Client Routing Deconstructor.** The SCRD decouples shared semantics from local biases, preserving fine-grained knowledge across heterogeneous clients while maintaining consistency through shared semantics, thereby mitigating knowledge contamination.

- **Poly-Consensus Aggregation.** We propose PCA, employing adaptive weighted averaging to ensure that the aggregated global LoRA parameters are aligned with the diverse distributions of participating clients, thus calibrating the global update direction.

- **Experimental Verification.** We fine-tune Llama2-7B (Touvron et al., 2023) , TinyLlama (Zhang et al., 2024b), Llama4-17B (Meta-AI, 2025) and Qwen3-8B (Yang et al., 2025) using RD-LoRA and evaluate on two benchmarks across three datasets. RD-LoRA outperforms state-of-the-art methods in both homogeneous and heterogeneous settings while relatively reducing communication overhead.

## 2 RELATED WORK

### 2.1 PARAMETER-EFFICIENT FINE-TUNING

LLMs such as GPT-3/4 (Brown et al., 2020; Achiam et al., 2023) and Llama-2/3 (Touvron et al., 2023; Grattafiori et al., 2024), are pre-trained on massive corpora and achieve strong performance across diverse applications (Raffel et al., 2020). To adapt them, instruction tuning (Mishra et al., 2021) and reinforcement learning with human feedback (Bai et al., 2022) are widely used, but they rely on centralised data and are impractical for resource-constrained devices given the large number of trainable parameters. PEFT (Ding et al., 2022; Hu et al., 2022) therefore updates only a small subset of parameters. Among additive PEFT methods, LoRA (Hu et al., 2022) inserts trainable low-rank adapters while freezing the backbone and has proved effective on complex tasks (Zhang et al., 2023b); notable variants include LongLoRA (Chen et al., 2023) for long-context efficiency and

AdaLoRA (Zhang et al., 2023a) for adaptive budget allocation. Owing to its simplicity, efficiency, and broad adoption, we focus on LoRA in this work.

## 2.2 FEDERATED FINE-TUNING HOMOGENEOUS LoRA

FL enables distributed fine-tuning of LLM, offering privacy protection and broad adoption in finance (Long et al., 2020; Chatterjee et al., 2023) and healthcare (Feng et al., 2022; Yan et al., 2024a; Jiang et al., 2023; Yan et al., 2024b). Tooling and benchmarks include FederatedScope-LLM for federated instruction tuning (Kuang et al., 2024) and OpenFedLLM for instruction tuning and value alignment (Ye et al., 2024), while FedIT establishes a LoRA-based PEFT baseline in FL (Zhang et al., 2024a). Methodologically, FedBiOT splits and compresses LLMs via bilevel optimisation to reduce cost (Wu et al., 2024), and FFA-LoRA highlights limitations of independently integrating LoRA matrices through specific A/B initialisation (Sun et al., 2024). In contrast, RD-LoRA employs an alternating-freeze framework to train all LoRA modules and mitigate aggregation noise, and further targets the often-overlooked heterogeneity of client resources.

## 2.3 FEDERATED FINE-TUNING HETEROGENEOUS LoRA

Non-IID heterogeneity (Li et al., 2019) arising from Dirichlet sampling and hardware differences motivates dynamic LoRA rank adjustment. Federated fine-tuning has explored heterogeneous ranks (Cho et al., 2024; Wang et al., 2024; Bai et al., 2024): Cho et al. (Cho et al., 2024) combine zero-padding with truncated aggregation to mix high- and low-rank LoRA, but padded zeros dilute strong-client signals and slow optimization; FLoRA (Wang et al., 2024) stacks client updates along the rank dimension to reconstruct full weights, which reduces noise yet incurs linear communication cost and limits flexibility; FlexLoRA (Bai et al., 2024) aggregates A and B separately and redistributes via SVD, adding computation and storage overhead, and introducing truncation error. LoRA-$A^2$ (Koo et al., 2024) alternates between freezing A and B and selects the salient parameters according to the importance of the data, reducing inconsistency and communication. However, it assumes a uniform global rank, whereas clients often begin with heterogeneous ranks.

## 3 METHODS

### 3.1 PRELIMINARIES

**Low-Rank Adaptation.** LoRA (Hu et al., 2022) reduces the number of trainable parameters by expressing the weight update as a low-rank factorization:

$$W = W_0 + \Delta W, \qquad \Delta W = BA. \tag{1}$$

Here $W, W_0, \Delta W \in \mathbb{R}^{d \times l}$, $B \in \mathbb{R}^{d \times r}$, $A \in \mathbb{R}^{r \times l}$, with $r \ll \min(d, l)$. We use $d$ for the output dimension, $l$ for the input dimension, and $r$ for the low rank.

**Aggregation Noise.** FedAvg (McMahan et al., 2017) updates the global model using a weighted average over $k$ clients. With LoRA, training/communicating only the low-rank factors $B$ and $A$ reduces the computation and bandwidth versus the full $\Delta W$. To define the aggregation problem with greater precision, we explicitly define two distinct terms: $\Delta W_{\text{Ideal}}$ for the ideal aggregation of the full updates, and $\Delta W_{\text{FedIT}}$ for the naive averaging of the low-rank factors separately used in FedIT (Zhang et al., 2024a). The non-equivalence between these two approaches is formally defined as:

$$\Delta W_{\text{FedIT}} = \left( \frac{1}{k} \sum_{i=1}^{k} B_i \right) \left( \frac{1}{k} \sum_{i=1}^{k} A_i \right) \neq \Delta W_{\text{Ideal}} = \frac{1}{k} \sum_{i=1}^{k} (B_i A_i) \tag{2}$$

To provide a deeper analysis of aggregation noise, we expand the expression for $\Delta W^{\text{FedIT}}$ based on the following decomposition:

$$\Delta W_{\text{FedIT}} = \left( \frac{1}{k} \sum_{i=1}^{k} B_i \right) \left( \frac{1}{k} \sum_{i=1}^{k} A_i \right) = \frac{1}{k} \underbrace{\left( \frac{1}{k} \sum_{i=1}^{k} (B_i A_i) \right)}_{\Delta W_{\text{Ideal}}} + \underbrace{\frac{1}{k^2} \sum_{i \neq j} (B_i A_j)}_{\text{aggregation noise}} \tag{3}$$

The cross-terms between the LoRA modules from different clients give rise to aggregation noise. Furthermore, by applying the $\frac{1}{k}$ scaling factor to both $B_i$ and $A_i$ independently, FedIT introduces an

erroneous $\frac{1}{k}$ coefficient to the ideal update component, $\Delta W_{\text{Ideal}}$, thereby exacerbating the error in the LoRA aggregation.

**Alternating Freezing Technique.** In (Koo et al., 2024), an alternative freezing scheme mitigates the noise of aggregation in LoRA-FL. In round $t$ ($B$-round), freeze the aggregated $\bar{A}^{(t)}$ and train only $B_i^{(t)}$: $\frac{1}{k} \sum_{i=1}^{k} B_i^{(t)} \bar{A}^{(t)}$. In round $t+1$ ($A$-round), freeze $\bar{B}^{(t+1)}$ and train only $A_i^{(t+1)}$: $\frac{1}{k} \sum_{i=1}^{k} \bar{B}^{(t+1)} A_i^{(t+1)}$. This alternation iteratively updates $A$ and $B$. However, in the presence of client-side data and rank heterogeneity, this approach does not adequately mitigate knowledge contamination or aggregation distortion. Consequently, we build on the alternating freeze training of $A$ and $B$ to better address these issues.

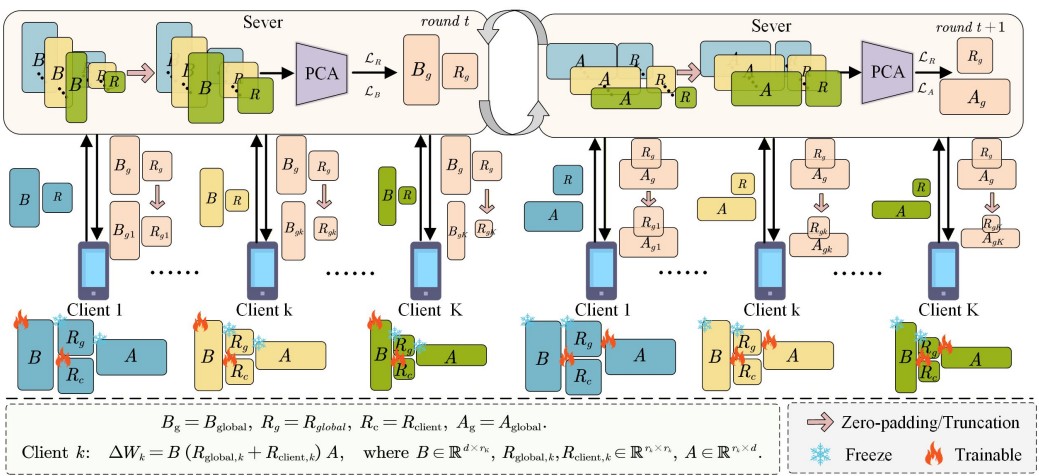

Figure 2: An overview of the proposed RD-LoRA. RD-LoRA alternates LoRA updates and introduces a deconstructed routing matrix $R = R_{\text{global}} + R_{\text{client}}$. In round $t$ ($B$-round) clients train $B$ and $R_{\text{client}}$ with $A$ frozen; in $t+1$ ($A$-round) they train $A$ and $R_{\text{client}}$ with $B$ frozen. Only the updated modules are transmitted. The server aggregates either $A$ or $B$, together with $R$, using PCA with adaptive position-wise weights and historical regularisation. This yields $A_{\text{global}}, B_{\text{global}}, R_{\text{global}}$ for redistribution. Heterogeneous ranks are handled by zero-padding or truncation to a common shape before aggregation and on client download. The detailed operations will be discussed in Section 3.

### 3.2 PROPOSED METHOD

In FL, clients hold complex non-IID data, and resource disparities further amplify biases in local updates. Even with LoRA, servers still face aggregation noise, knowledge contamination, and aggregation distortion when merging low-rank matrices. To address these challenges, we propose Routing-Deconstructed Low-Rank Adaptation (RD-LoRA) that reduces server communication overhead while effectively suppressing noise and mitigating contamination and distortion, as illustrated in Figure 2. To address knowledge contamination, we introduce Server-Client Routing Deconstructor (SCRD), which separates learning into two paths: a global consensus route for shared semantics and a local private route for device-specific knowledge. Freezing the consensus route while training only the private route prevents gradient interference and preserves local features. To reduce aggregation distortion, we propose Poly-Consensus Aggregation (PCA), which combines additive consensus with attention-based adaptive weighting for fine-grained fusion of heterogeneous low-rank matrices. An $L_2$ proximal regulariser further constrains global updates near local optima, ensuring balanced consistency and local optimality.

**Server-Client Routing Deconstructor (SCRD)**

Departing from Equation 1, we introduce an SCRD to separate shared representations from local bias and mitigate knowledge contamination in realistic FL. Concretely, we place $R = R_{\text{client}} + R_{\text{global}}$ between the feature projection $B$ and the low-rank direction $A$, with $R, R_{\text{client}}, R_{\text{global}} \in \mathbb{R}^{r \times r}$, so that the weight update as:

$$\triangle W = BRA = B \left( R_{\text{client}} + R_{\text{global}} \right) A, \tag{4}$$

where $R$ acts as a connector that routes between the low-rank direction $A$ and the feature projection $B$, shaping interactions in the low-rank subspace. To retain global information while exploiting client signals, we decompose $R = R_{\text{client}} + R_{\text{global}}$. Here, $R_{\text{client}}$ is initialised and updated on the client, captures per-round local changes, and participates in both forward and backward passes. In contrast, $R_{\text{global}}$ is aggregated on the server, encodes cross-client shared directions, and is used only in the client forward pass, which helps mitigate knowledge contamination. Thus, $R$ fuses global and client-specific knowledge, preserving consistency in FL while adapting to non-IID data.

In practical FL, pronounced non-IID means that naively adding local and global updates can let global semantics dominate, drowning out local features and worsening client drift. If $R_{\text{global}}$ is overweighted, it governs low-rank routing in the forward pass and limits $R_{\text{client}}$ from expressing client-specific structure. We therefore down-weight the global term with a scaling coefficient $\gamma$, giving greater emphasis to local information:

$$\triangle W = B\left(R_{\text{client}} + \gamma R_{\text{global}}\right) A. \tag{5}$$

In this setting, $R_{\text{global}}$ serves as a weak anchor, providing stable directional guidance without inhibiting local updates, thereby enabling both the client models and the subsequently aggregated model to adapt to heterogeneous local data whilst preserving global consistency. We set $\gamma = 0.3$ in all experiments presented in the paper. *Please see experiments on $\gamma$ in Supplementary Material A.*

During client-side training, we adopt an alternating freezing strategy to reduce aggregation noise and knowledge contamination along the multiplicative chain $BRA$. In $B$-round, only $B$ and $R_{\text{client}}$ are trained, whilst $A$ is frozen: $\frac{\partial \mathcal{L}}{\partial B} \neq 0, \frac{\partial \mathcal{L}}{\partial A} = 0, \frac{\partial \mathcal{L}}{\partial R_{\text{client}}} \neq 0$. In $A$-round, only A and $R_{\text{client}}$ are trained, whilst $B$ is frozen: $\frac{\partial \mathcal{L}}{\partial A} \neq 0, \frac{\partial \mathcal{L}}{\partial B} = 0, \frac{\partial \mathcal{L}}{\partial R_{\text{client}}} \neq 0$. In addition, apply gradient shielding constraints to routing anchors: $\frac{\partial \mathcal{L}}{\partial R_{\text{global}}} = 0$.

To stabilize the global direction, we insulate $R_{\text{global}}$ from short-term local gradients, while $R_{\text{client}}$ captures a fine-grained client-specific bias. In round $t$, the local objective of client $i$ is formulated:

$$\mathcal{L}_i^{(t)} = \mathbb{E}_{(x,y)\sim\mathcal{D}_i}\left[\ell\left(f\left(x; \{W_0^\ell + B_i^{\ell,(t)}\left(R_{\text{client},i}^{\ell,(t)} + \gamma R_{\text{global},i}^{\ell,(t)}\right)A_i^{\ell,(t)}\}_{\ell\in\mathcal{L}}\right), y\right)\right]. \tag{6}$$

Let $D_i$ denote client $i$'s data distribution, $\mathcal{L}$ the set of layers into which LoRA is injected, and $\ell(\cdot, \cdot)$ the task loss. According to Equation 6, local adaptation uses $R_{\text{client}}$ together with the round's trainable factor ($A$ or $B$), whilst global consistency is anchored by the forward-injected $\gamma R_{\text{global}}$. Communication only uploads the trained part. In round $t$ ($B$-round), each client uploads $\{B_i^{(t)}, R_{\text{client},i}^{(t)}\}$; in round $t+1$ ($A$-round), they upload $\{A_i^{(t+1)}, R_{\text{client},i}^{(t+1)}\}$. In each round, the server aggregates these to obtain either $\{B_{\text{global},i}^{(t)}, R_{\text{global},i}^{(t)}\}$ or $\{A_{\text{global},i}^{(t+1)}, R_{\text{global},i}^{(t+1)}\}$, which are then broadcast back to clients. The SCRD scheme effectively mitigates knowledge contamination and balances global consistency with local adaptability at no additional cost.

**Theoretical Validation.** By decomposing the routing matrix into a frozen global anchor $R_{\text{global}}$ and a trainable local component $R_{\text{client}}$, RD-LoRA imposes stricter sensitivity bounds on LoRA submodules. Compared to jointly training $R_{\text{client}} + R_{\text{global}}$, updating only $R_{\text{client}}$ reduces gradient magnitude, constrains updates under a fixed step size, and enables local adaptation while mitigating interference with globally consistent knowledge. The following theorem captures this effect.

***Theorem 3.1 (Stricter update bounds via deconstructed routing).*** *Let $R = R_{\text{client}} + R_{\text{global}}$, where $R_{\text{client}}$ is trainable and $R_{\text{global}}$ is either frozen or trainable. Assume that the symmetric part of $R_{\text{global}}^\top R_{\text{client}}$ is positive definite and $BG^\top \neq 0$, $G^\top A \neq 0$, where $G = \frac{\partial y}{\partial W}$. Let $y'$ be the output with frozen $R_{\text{global}}$ and trainable $R_{\text{client}}$, and $y$ be the output when both are jointly trained. Then:*

$$\left\|\frac{\partial y'}{\partial A}\right\|_F^2 - \left\|\frac{\partial y}{\partial A}\right\|_F^2 < 0, \quad \left\|\frac{\partial y'}{\partial B}\right\|_F^2 - \left\|\frac{\partial y}{\partial B}\right\|_F^2 < 0. \tag{7}$$

***Proof.*** *For brevity write $R_{\text{g}} = R_{\text{global}}$ and $R_{\text{c}} = R_{\text{client}}$.*

***Sensitivity with respect to*** *$A$. We have:*

$$\frac{\partial y'}{\partial A} = G\, B^\top R_{\text{c}}^\top, \quad \frac{\partial y}{\partial A} = G\, B^\top (R_{\text{g}} + R_{\text{c}})^\top. \tag{8}$$

*Using $\|X\|_F^2 = \mathrm{Tr}(X^\top X)$, we have:*

$$\left\|\frac{\partial y'}{\partial A}\right\|_F^2 - \left\|\frac{\partial y}{\partial A}\right\|_F^2 = \mathrm{Tr}\left[R_\mathrm{c}\, BG^\top GB^\top\, R_\mathrm{c}^\top\right] - \mathrm{Tr}\left[(R_\mathrm{g} + R_\mathrm{c})\, BG^\top GB^\top\, (R_\mathrm{g} + R_\mathrm{c})^\top\right]$$

$$= -\mathrm{Tr}\left[\left(R_\mathrm{g}^\top R_\mathrm{g} + 2R_\mathrm{g}^\top R_\mathrm{c}\right)\left(BG^\top GB^\top\right)\right] < 0. \tag{9}$$

*Here, the inequality holds because the symmetric part of $R_\mathrm{g}^\top R_\mathrm{c}$ is positive definite and $BG^\top \neq 0$.*

***Sensitivity with respect to $B$.** Similarly:*

$$\frac{\partial y}{\partial B} = (R_\mathrm{g} + R_\mathrm{c})^\top A^\top G, \qquad \frac{\partial y'}{\partial B} = R_\mathrm{c}^\top A^\top G. \tag{10}$$

*Using $\|X\|_F^2 = \mathrm{Tr}(X^\top X)$, we have:*

$$\left\|\frac{\partial y'}{\partial B}\right\|_F^2 - \left\|\frac{\partial y}{\partial B}\right\|_F^2 = \mathrm{Tr}\left[R_\mathrm{c}^\top A^\top GG^\top AR_\mathrm{c}\right] - \mathrm{Tr}\left[(R_\mathrm{g} + R_\mathrm{c})^\top A^\top GG^\top A(R_\mathrm{g} + R_\mathrm{c})\right]$$

$$= -\mathrm{Tr}\left[\left(R_\mathrm{g}^\top R_\mathrm{g} + 2R_\mathrm{c}^\top R_\mathrm{g}\right)\left(A^\top GG^\top A\right)\right] < 0. \tag{11}$$

*Here, the inequality holds because the symmetric part of $R_\mathrm{g}^\top R_\mathrm{c}$ is positive definite and $G^\top A \neq 0$. An explanation regarding the condition that the symmetric part of $R_\mathrm{global}^\top R_\mathrm{client}$ is positive definite can be found in Supplementary Material B. Furthermore, Supplementary Material C demonstrates the effectiveness of SCRD in ensuring the separation of local and global knowledge on the server side.*

### Poly-Consensus Aggregation (PCA)

FedAvg is computationally simple but, under non-IID data, it induces aggregation distortion. Clients contribute updates with unequal reliability across parameter locations/subspaces, so naively averaging $A$, $B$ and $R$ smears informative signals and causes the global model to drift from each client's neighborhood optimum. We therefore adopt PCA mechanism, a server-

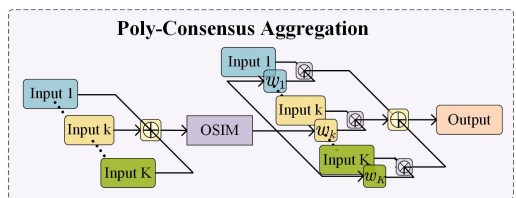

Figure 3: Overview of the proposed PCA.

side, fine-grained, history-regularized fusion of LoRA submodules. For each $\theta \in \{A \text{ or } B,\, R\}$, let $I_t^\theta$ denote the set of clients participating in round $t$, and represent their uploads as matrices $\{F_i \in \mathbb{R}^{d_1 \times d_2} \mid i \in I_t^\theta\}$, where each $F_i$ corresponds to a LoRA submodule ($A$ or $B$, $R$) and serves as the basic unit of aggregation in PCA. With rank heterogeneity, the matrices are padded or truncated to the maximum shape $(d_1, d_2)$ before upload and reshaped to the local form after aggregation. As illustrated in Figure 3, the proposed PCA framework operates in three main stages. First, it constructs a cross-client contextual prior. It then uses an Omni-Scale Integration Module (OSIM) to assign position-wise attention weights across clients and select the most reliable contributor at each location. Next, it performs weighted fusion with optional refinement for robust, non-linear aggregation. Finally, it applies historical steady alignment with element-wise $L_2$ regularization. This keeps global updates within a stable region between client solutions and the previous global state, preserving consistency and local optimality. The details are as follows.

**Step 1: Poly-Fusion Gate.** To prevent pre-aggregation "mean smearing", we first construct a cross-client contextual prior via lossless synthesis.

$$F_0 = \sum_{i \in I_t^\theta} F_i. \tag{12}$$

This is not a global estimate; it supplies global statistics for subsequent attention, revealing the merged structure and reducing the smearing of weak yet informative local patterns.

**Step 2: Omni-Scale Integration Module (OSIM).** Given the client-specific submatrices $\{F_i\}_{i \in I_t^\theta}$ and the poly-fusion map $F_0$, OSIM assigns *position-dependent* aggregation weights to each client. For every client $i$ and entry $(u, v)$, we first build a local descriptor:

$$z_i[u, v] = \left[\, F_i[u, v],\ F_0[u, v],\ F_i[u, v] - F_0[u, v]\,\right]^\top \in \mathbb{R}^3, \tag{13}$$

which encodes the client update, the cross-client prior, and their deviation at that location. This descriptor is passed through a lightweight, shared two-layer MLP:

$$h_i[u,v] = \sigma\big(\mathcal{W}_1 z_i[u,v] + \beta_1\big), \qquad s_i[u,v] = \mathcal{W}_2^\top h_i[u,v] + \beta_2, \tag{14}$$

where $\mathcal{W}_1, \beta_1, \mathcal{W}_2, \beta_2$ are OSIM parameters and $\sigma(\cdot)$ is a point-wise nonlinearity. The resulting logits $s_i[u,v]$ are then normalized over the client dimension via a temperature-scaled softmax:

$$w_i[u,v] = \frac{\exp\big(s_i[u,v]/\tau\big)}{\sum_{j \in I_t^\theta} \exp\big(s_j[u,v]/\tau\big)}, \quad \sum_i w_i[u,v] = 1, \quad w_i[u,v] \in [0,1]. \tag{15}$$

OSIM thus computes position-wise weights $w_i[u,v]$ that favor more reliable client updates at each parameter entry. This fine-grained, adaptive weighting is crucial for non-IID settings, as it acknowledges that the optimal contributor can vary from one parameter position to another.

**Step 3: Weighted fusion with optional iterative refinement.** Given position-wise normalized weights $\{w_i\}$, compute $F_1 = \sum_{i \in I_t^\theta} w_i \odot F_i$, where $\odot$ denotes element-wise multiplication. For added robustness beyond a linear blend, optionally re-estimate weights conditioned on $F_1$ to obtain $\{w_i'\}$ and compute $F_{\text{final}} = \sum_{i \in I_t^\theta} w_i' \odot F_i$. This second pass induces data-dependent nonlinearity.

**Step 4: Historical steady alignment.** Historical steady alignment addresses between-round oscillation and update-direction swings by adding an element-wise $L_2$ historical constraint that keeps the new global estimate within a steady region between a convex combination of current client solutions and the previous global solution. The objective is:

$$\mathcal{L}_\theta^{(t)}(\Theta) = \underbrace{\sum_{i \in I_t^\theta} \| W_i \odot (\Theta - \theta_i) \|_F^2}_{\text{close to current trusted solutions}} + \underbrace{\lambda_\theta \left\| \Theta - \theta_{\text{global}}^{(t-1)} \right\|_F^2}_{\text{anchored to last global}}, \tag{16}$$

where the weighted Frobenius term is:

$$\left\| \Theta - \theta_i \right\|_{F,(w)}^2 = \sum_{u,v} w_i[u,v] \left( \Theta[u,v] - \theta_i[u,v] \right)^2. \tag{17}$$

Here $\theta_i = F_i$, $\theta_{\text{global}}^{(t-1)}$ is the previous global parameter of the same type, and $\lambda_\theta \geq 0$ controls the strength of the steady-state anchor (often $\lambda_R \geq \lambda_A, \lambda_B$ to stabilise the routing anchor). This aggregation admits a closed-form solution for each element:

$$\Theta^{(t)}[u,v] = \frac{\sum_i w_i[u,v]\, \theta_i[u,v] + \lambda_\theta\, \theta_{\text{global}}^{(t-1)}[u,v]}{\sum_i w_i[u,v] + \lambda_\theta}. \tag{18}$$

Setting $\Theta \in \{A \text{ or } B, R\}$ yields $B_{\text{global}}^{(t)}, R_{\text{global}}^{(t)}$ or $A_{\text{global}}^{(t+1)}, R_{\text{global}}^{(t+1)}$, which serve as initialisations for the subsequent round. *Please see the overall algorithm of RD-LoRA in Supplementary Material D.*

## 4 EXPERIMENTS

### 4.1 EXPERIMENT SETUP

**Comparison methods.** We benchmark RD-LoRA against five state-of-the-art methods for federated fine-tuning of heterogeneous LoRA: FFA-LoRA (Sun et al., 2024), Zero-padding (FedIT) (Cho et al., 2024), FLoRA (Wang et al., 2024), FlexLoRA (Bai et al., 2024) and LoRA-A$^2$ (Koo et al., 2024), plus a Local baseline. FFA-LoRA (Sun et al., 2024) is included only for homogeneous ranks, as it assumes identical ranks and is not applicable to heterogeneous LoRA. For fairness, all methods are reproduced in OpenFedLLM (Ye et al., 2024) with the same backbone, data splits, and training settings. *Please see the details about the procedures in Supplementary Material E.*

**Datasets and evaluation.** We train RD-LoRA on three datasets: Alpaca-GPT4 (Peng et al., 2023) (52k GPT-4–generated instruction–response pairs based on Alpaca (Wang et al., 2022b)), Databricks-dolly-15k (Zhang et al., 2024a) (diverse instruction tasks), Wizard (Luo et al., 2023) (70k instruction–output pairs), and Big-Math (Albalak et al., 2025) (a collection of over 250,000 high-quality math problems). For federated training, data are partitioned with a Dirichlet distribution ($\alpha = 0.5$). We further evaluate on closed benchmarks and open benchmarks, using MMLU (Hendrycks et al.,

| Method (Llama2) | MMLU | | | MT-Bench |
| Homo \| Heter | Wizard | Dolly | Alpaca | Alpaca |
|---|---|---|---|---|
| Zero-shot | | 35.09±0.00 | | 3.730±0.000 |
| Local | 33.11±0.33 \| 32.63±0.38 | 40.02±0.53 \| 39.11±0.57 | 32.10±0.23 \| 31.94±0.28 | 2.934±0.032 \| 2.851±0.030 |
| FFA-LoRA (Sun et al., 2024) | 34.68±0.27 \| 34.33±0.31 | 44.32±0.51 \| 43.55±0.56 | 32.58±0.11 \| 31.74±0.27 | 3.606±0.037 \| 3.584±0.038 |
| Zero-padding (Cho et al., 2024) | 34.73±0.32 \| 34.47±0.40 | 44.11±0.43 \| 43.77±0.52 | 32.87±0.20 \| 32.26±0.36 | 3.622±0.036 \| 3.569±0.034 |
| FLoRA (Wang et al., 2024) | 35.78±0.26 \| 35.44±0.34 | 45.27±0.57 \| 44.94±0.47 | 35.11±0.13 \| 34.57±0.22 | 3.874±0.026 \| 3.801±0.034 |
| LoRA-A$^2$ (Koo et al., 2024) | 35.81±0.31 \| 35.64±0.33 | 45.69±0.23 \| 45.38±0.27 | 35.71±0.24 \| 35.62±0.31 | 4.043±0.031 \| 4.026±0.035 |
| FlexLoRA (Bai et al., 2024) | 36.07±0.18 \| 36.17±0.19 | 45.72±0.34 \| 45.91±0.38 | 35.83±0.18 \| 36.20±0.24 | 4.102±0.025 \| 4.131±0.026 |
| RD-LoRA (Ours) | **36.68±0.13 \| 36.54±0.17** | **47.70±0.31 \| 47.16±0.26** | **37.58±0.20 \| 37.21±0.22** | **4.224±0.021 \| 4.205±0.024** |

Table 1: Performance comparison of RD-LoRA with baseline methods on MMLU and MT-Bench using Llama2-7B. The results distinguish between Homogeneous (left) and Heterogeneous (right) LoRA rank settings.

2020) for the former and MT-Bench (Zheng et al., 2023) for the latter. Closed tasks have definitive answers enabling objective automated scoring, so MMLU probes factual knowledge and verifiable reasoning; open tasks permit free-form generation with multiple valid responses, so MT-Bench assesses conversational quality via model or human judgements. Additionally, the AIME25 benchmark (AIME, 2025) is specifically used to assess the mathematical reasoning capabilities developed through the Big-Math dataset.

**Training details.** We use the pre-trained Llama2-7B (7 billion parameters) (Touvron et al., 2023), TinyLlama (1.1 billion parameters) (Zhang et al., 2024b), Llama4-17B (17 billion parameters) (Meta-AI, 2025) and Qwen3-8B (8 billion parameters) (Yang et al., 2025) as base models and ran 200 rounds of federated communication. The learning rate starts at 5e-5 and follows a cosine schedule to 1e-6 by the end of 200 rounds. In each round, two clients are sampled at random, and each trains for 10 epochs with a batch size of 16. (1) Homogeneous LoRA: the rank is 16 following FLoRA. (2) Heterogeneous LoRA: following FLoRA's real-world simulation, the ten clients use local ranks [64, 32, 16, 16, 8, 8, 4, 4, 4, 4].

## 4.2 COMPARISON EXPERIMENT

**Homogeneous LoRA.** In the homogeneous LoRA setting, all clients used the same LoRA rank of 16. Based on Table 1 Homo and Table 2a, we observe the following: (1) All six fine-tuning methods outperform local training. On the Dolly-15k dataset, RD-LoRA improves MMLU performance by 7.68 for Llama2 and 7.31 for TinyLlama over the local baseline. (2) Zero-padding and FFA-LoRA perform relatively poorly. The former suffers from aggregation noise, while the latter restricts optimisation by freezing the A module. Notably, in this setting, zero-padding reduces to simply aggregating B and A modules on the server. (3) FlexLoRA outperforms FLoRA. Both aggregate the matrix product of B and A, but FlexLoRA applies SVD to the aggregated weights and sends a new module to clients. FLoRA merges the weights with the local LLM and reinitialises training, which may destabilise learning. (4) LoRA-A$^2$ mitigates aggregation noise by alternately freezing B and A, outperforming three other methods apart from FlexLoRA, while incurring lower communication cost. (5) RD-LoRA effectively addresses aggregation noise, distortion, and knowledge contamination, consistently outperforming all baselines with relatively low communication overhead.

**Heterogeneous LoRA.** Heterogeneous settings as described in Section 4.1. In Table 1 Heter and Table 2b, most methods show modest degradation under heterogeneous ranks relative to homogeneous ones. On Llama2, the maximum MMLU drops for Zero-padding, FLoRA, LoRA-A$^2$ and RD-LoRA are 0.61, 0.54, 0.34 and 0.31; TinyLlama shows more moderate variation, with maximum drops of 0.45, 0.41, 0.54, and 0.15, slightly smaller than for Llama2. This suggests that heterogeneity-induced degradation is better controlled on the smaller model under RD-LoRA. Under heterogeneity, FlexLoRA performs better than in the homogeneous configuration, evidencing effective handling of resource heterogeneity and improved utilisation. LoRA-A$^2$ continues to surpass FLoRA and Zero-padding by suppressing aggregation noise. RD-LoRA mitigates aggregation noise, distortion and knowledge contamination. As a result, It remains optimal under heterogeneous settings with both base models, and the performance drop is relatively controlled. On larger models, as shown in Table 3, while baselines like Zero-padding (FedIT) suffer from performance degradation in heterogeneous settings (dropping below Zero-shot levels), our method consistently outperforms FlexLoRA. Specifically, RD-LoRA achieves gains of 0.97 and 1.27 on Qwen3-8B, and 1.43 and 1.18 on Llama4-17B for the Dolly and BigMath tasks, respectively.

| Method | MMLU | | | MT-Bench |
| --- | --- | --- | --- | --- |
| | Wizard | Dolly | Alpaca | Alpaca |
| Zero-shot | | 45.28 | | 4.351 |
| Local | 47.62 | 55.46 | 42.88 | 3.567 |
| FFA-LoRA (Sun et al., 2024) | 49.17 | 58.35 | 44.35 | 4.322 |
| Zero-padding (Cho et al., 2024) | 49.33 | 58.80 | 44.67 | 4.349 |
| FLoRA (Wang et al., 2024) | 50.14 | 59.32 | 46.04 | 4.400 |
| LoRA-$A^2$ (Koo et al., 2024) | 50.66 | 60.07 | 46.75 | 4.503 |
| FlexLoRA (Bai et al., 2024) | 51.07 | 60.77 | 47.22 | 4.578 |
| RD-LoRA (Ours) | **52.26** | **62.61** | **48.71** | **4.744** |

(a) Homogeneous LoRA ranks.

| Method | MMLU | | | MT-Bench |
| --- | --- | --- | --- | --- |
| | Wizard | Dolly | Alpaca | Alpaca |
| Zero-shot | | 45.28 | | 4.351 |
| Local | 47.43 | 55.08 | 42.62 | 3.514 |
| FFA-LoRA (Sun et al., 2024) | 49.05 | 58.11 | 44.23 | 4.266 |
| Zero-padding (Cho et al., 2024) | 49.20 | 58.66 | 44.22 | 4.308 |
| FLoRA (Wang et al., 2024) | 49.73 | 59.04 | 45.87 | 4.372 |
| LoRA-$A^2$ (Koo et al., 2024) | 50.12 | 59.87 | 46.51 | 4.453 |
| FlexLoRA (Bai et al., 2024) | 51.31 | 61.08 | 47.56 | 4.598 |
| RD-LoRA (Ours) | **51.89** | **62.46** | **48.54** | **4.715** |

(b) Heterogeneous LoRA ranks.

Table 2: Performance comparison of RD-LoRA with baseline methods on MMLU and MT-Bench using TinyLlama-1.1B under heterogeneous (left) and homogeneous (right) data settings.

| Method | MMLU | AIME25 |
| --- | --- | --- |
| | Dolly | Big-Math |
| Zero-shot | 72.49 | 17.11 |
| Zero-padding (Cho et al., 2024) | 71.09 | 16.07 |
| FlexLoRA (Bai et al., 2024) | 73.14 | 18.46 |
| RD-LoRA (Ours) | **74.11** | **19.73** |

(a) Heterogeneous LoRA ranks (Qwen3-8B).

| Method | MMLU | AIME25 |
| --- | --- | --- |
| | Dolly | Big-Math |
| Zero-shot | 75.13 | 9.02 |
| Zero-padding (Cho et al., 2024) | 73.64 | 8.85 |
| FlexLoRA (Bai et al., 2024) | 75.67 | 11.03 |
| RD-LoRA (Ours) | **77.10** | **12.21** |

(b) Heterogeneous LoRA ranks (Llama4-17B).

Table 3: Performance comparison of RD-LoRA with baseline methods on MMLU and AIME25 benchmarks under heterogeneous LoRA-rank settings, utilizing Qwen3-8B and Llama4-17B as base models.

**Communication overhead.** Communication cost is critical in real-world FL. We therefore compare single-round upload and download costs across methods, with transmission parameters normalized for fairness. As shown in Figure 5, FFA-LoRA has the lowest cost, as it uploads only module B each round. Our method and LoRA-$A^2$ also keep overhead low by uploading only module A or B; our method additionally transmits a routing matrix. FlexLoRA and Zero-padding require both modules to be uploaded and downloaded every round. FLoRA incurs the highest cost because products B and A must be downloaded. These results highlight the benefit of alternate freezing and show that our method achieves a favorable balance between performance and communication efficiency. *For a detailed comparison of computational complexity, please refer to the supplementary material F.*

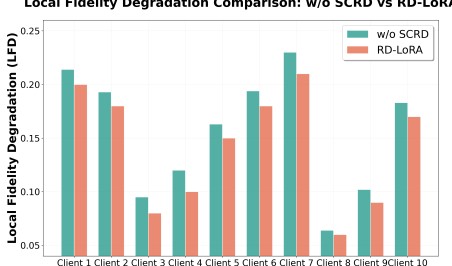

Figure 4: LFD shows RD-LoRA's lower fidelity drop, indicating better local knowledge retention.

Figure 5: Communication cost comparison. RD-LoRA achieves relatively low overhead.

### 4.3 ABLATION EXPERIMENT

To verify the effectiveness of the SCRD mechanism and the PCA mechanism in RD-LoRA, we performed several ablation experiments using the Llama2-7B base model. In particular, "w/o SCRD" refers to training only module B or module A of the clients in the heterogeneous setting, with the update expressed as $W = BA$. "w/o PCA" denotes replacing the PCA strategy with the weighted average aggregation of FedAvg in the heterogeneous setting. "w/o OSIM" refers to the model lacking the per-position attention mechanism, while "w/o Hist. Align." indicates the model without the historical steady alignment term ($\lambda$).

**The effects of SCRD.** To assess the effectiveness of SCRD, we compared the Local Fidelity Degradation (LFD) of clients under w/o SCRD and RD-LoRA, as shown in Figure 4. LFD measures the loss difference on the local validation set before and after fine-tuning, where lower values indicate better retention of local knowledge. As illustrated in Figure 4, RD-LoRA consistently achieves lower LFD values than w/o SCRD across almost all clients, with the advantage being particularly

pronounced for Client 3, Client 4, and Client 7. This demonstrates that SCRD better preserves local knowledge, reduces the risk of losing client-specific information, and effectively alleviates knowledge pollution. In Table 4, the performance of w/o SCRD is at least 0.63 percentage points lower than that of RD-LoRA; whereas in Table 1 Heter, w/o PCA achieves overall performance on MMLU that even surpasses FlexLoRA, with a maximum lead of 0.35 percentage points. These results provide further confirms that SCRD not only mitigates knowledge pollution but also enhances the overall capability of the global model across different datasets.

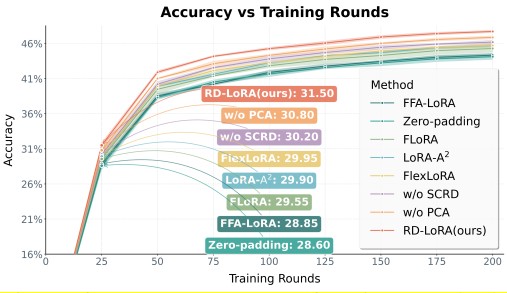

Figure 6: Accuracy Curves of Various Methods across Training Epochs.

| Method | MMLU | | |
|---|---|---|---|
| Heter | Wizard | Alpaca | Dolly |
| w/o SCRD | 35.91 | 45.88 | 36.11 |
| w/o PCA | 36.23 | 46.26 | 36.35 |
| w/o OSIM | 36.28 | 46.31 | 36.37 |
| w/o Hist. Align. | 36.41 | 46.74 | 36.66 |
| RD-LoRA (full pipeline) | **36.54** | **47.16** | **37.21** |

Table 4: Ablation study of RD-LoRA. "Heter" denotes settings with heterogeneous LoRA ranks.

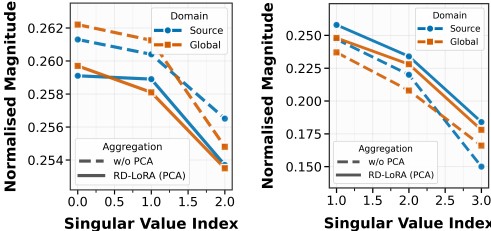

(a) With RD-LoRA.  (b) Without PCA.

Figure 7: Cosine similarity visualization of the local model processed by w/o PCA or RD-LoRA with the global model.

Figure 8: Singular value distributions (via SVD) for the source LoRA and its transforms (w/o PCA, RD-LoRA). Left: round 5; Right: round 9.

**The effects of PCA.** To assess the effect of this mechanism, we visualize the cosine similarity between clients and between each client and the global model, together with singular value distributions before and after aggregation. As shown in Figure 7, RD-LoRA achieves a higher similarity than w/o PCA, indicating that the PCA mechanism aligns the global direction more evenly between clients, retains greater local information and mitigates distortion of aggregation. To highlight the differences before and after aggregation, we compared the normalized singular value distributions, as shown in Figure 8. PCA better preserves clients' principal information, keeping distributions closer to the original LoRA, while FedAvg diverges. In Table 4, underperforms RD-LoRA in all cases ($\geq 0.31$), confirming that PCA reduces information loss and aggregation distortion and improves global performance. Both OSIM and Historical Alignment are essential; removing either degrades the performance of the full RD-LoRA model. On the Alpaca dataset, the full model achieves a score of 37.21, whereas removing Historical Alignment drops it to 36.66, and removing OSIM lowers it further to 36.37.

Figure 6 presents 200-round test accuracy curves for RD-LoRA and all baselines on the Dolly dataset. Curves show the mean of five runs, with shaded areas indicating standard deviation. The results demonstrate RD-LoRA's superior convergence, consistently outperforming the strongest baseline, FlexLoRA, from round 25 onwards.

# 5 CONCLUSION

We present RD-LoRA for heterogeneous federated fine-tuning. It builds on an existing alternating freezing strategy to suppress aggregation noise while reducing communication cost. To address the remaining challenges, the Server-Client Routing Deconstructor (SCRD) mitigates knowledge contamination by disentangling shared semantics from client biases, and the Poly-Consensus Aggregation (PCA) alleviates aggregation distortion through adaptive weighting with historical regularization. On Llama2 and TinyLlama, across three benchmarks and two tasks, RD-LoRA surpasses state-of-the-art baselines under both homogeneous and heterogeneous settings while maintaining low communication overhead, balancing the trade-off of performance and communication.

## ETHICS STATEMENT

This work adheres to the ICLR Code of Ethics . Our study does not involve human subjects, personal data, or sensitive demographic attributes, and thus raises no concerns related to privacy, security, or fairness. All datasets employed in our experiments (Alpaca-GPT4, Dolly-15k, and Wizard) are publicly available, widely adopted benchmarks in the research community, and we clearly describe their usage and preprocessing procedures in the supplementary materials. We carefully considered the potential risks of misuse: while RD-LoRA improves the efficiency and robustness of federated fine-tuning, it does not introduce capabilities that could reasonably be expected to cause societal harm beyond those already present in existing PEFT–FL methods. No undisclosed conflicts of interest or external sponsorship have influenced this research. We are committed to principles of research integrity, including transparency, reproducibility, and responsible reporting, and have made all methodological details available to enable independent verification.

## REPRODUCIBILITY STATEMENT

We have made extensive efforts to ensure the reproducibility of our work. The main text provides a detailed description of the proposed RD-LoRA framework, including algorithmic steps, theoretical analysis, and training procedures. Hyperparameter configurations, model architectures, and experimental setups are documented in Section 4 (Experiments), with additional implementation details provided in Appendix B (Comparative Methodology) and Appendix C (Effect of $\gamma$). The full proofs of the theoretical results are included in Section 3.2 and the supplementary materials. For datasets (Alpaca-GPT4, Dolly-15k, Wizard), partitioning strategies (Dirichlet $\alpha = 0.5$), and evaluation protocols (MMLU and MT-Bench) are described in Section 4.1 and the supplementary materials. These resources collectively enable independent researchers to reproduce our results and validate our claims.

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

## A  EFFECT OF $\gamma$ ON PERFORMANCE

Table 5 presents the MMLU performance of Wizard, Alpaca, and Dolly under varying values of the routing balance factor $\gamma$, which controls the relative contribution of the global routing anchor $R_{\text{global}}$ during client-side training. Across all three models, performance remains stable when $\gamma$ lies between 0.1 and 0.5. Notably, the optimal values appear around $\gamma = 0.3$ for Wizard and Dolly, and $\gamma = 0.2$ for Alpaca. These results suggest that moderate integration of the global anchor improves generalisation without overwhelming local adaptation. Excessively high values of $\gamma$ may dilute client-specific information, whereas overly low values may lead to insufficient global guidance.

| MMLU | $\gamma = 0.1$ | $\gamma = 0.2$ | $\gamma = 0.3$ | $\gamma = 0.4$ | $\gamma = 0.5$ |
|---|---|---|---|---|---|
| Wizard | 36.49 | 36.48 | **36.54** | 36.45 | 36.43 |
| Alpaca | 47.12 | **47.18** | 47.16 | 47.14 | 47.11 |
| Dolly | 37.13 | 37.16 | **37.21** | 37.15 | 37.12 |

Table 5: Results of Wizard, Alpaca, and Dolly on MMLU under different $\gamma$ values.

## B  JUSTIFICATION FOR $R_{global}^T R_{client}$

We contend that this assumption is realistic as it is a natural consequence of our specific training dynamic, where $R_{client}$ acts as a residual correction to the frozen anchor $R_{global}$.

### B.1  CONCISE EXPLANATION

The assumption holds because $R_{client}$ is initialized to zero and trained to minimize local loss in the context of a frozen $R_{global}$. To effectively reduce loss, the optimizer inherently learns an $R_{client}$ that creates constructive interference with the global anchor. Learning an antagonistic $R_{client}$ (one that cancels out the global prior) would be a sub-optimal optimization path. Therefore, the assumption that the symmetric part of $R_{global}^T R_{client}$ is positive definite is simply the mathematical formalization of this expected cooperative alignment.

### B.2  FORMAL JUSTIFICATION

(1) Constructive Alignment Condition:

Since $R_{client}$ is learned as a residual to improve the representation provided by $R_{global}$, the transformation applied by $R_{client}$ should be synergistic with $R_{global}$ for a typical feature vector $x$. Geometrically, this implies that the update direction should align constructively with the anchor:

$$\langle R_{global}x, R_{client}x \rangle > 0. \quad (E.2.1)$$

(2) Matrix Formulation:

Expanding the inner product yields:

$$(R_{global}x)^T(R_{client}x) = x^T R_{global}^T R_{client}x > 0. \quad (E.2.2)$$

(3) Connection to Positive Definiteness:

The condition $x^T M x > 0$ for all non-zero $x$ is the definition of positive definiteness for the symmetric part of a matrix $M$. Thus, the cooperative nature of the residual learning implies:

$$\text{Sym}(R_{global}^T R_{client}) > 0. \quad (E.2.3)$$

This confirms that our assumption is a direct mathematical consequence of the optimizer seeking a constructive solution within our residual learning framework.

# C   THEOREM 3.2 (COMPARATIVE ANALYSIS OF GLOBAL ROUTING MATRIX CONVERGENCE)

Assume the ideal optimal routing matrix $R_i^*$ for each client $i$ decomposes into a shared knowledge core $R_{shared}^*$ and a zero-mean, client-specific bias $\delta_i$ (i.e., $R_i^* = R_{shared}^* + \delta_i$, with $\mathbb{E}_i[\delta_i] = 0$).

We compare two scenarios for updating the global routing anchor $R_{global}^{(t)}$:

**Scenario A:** RD-LoRA (Client trains $R_{client,i}$ and uploads $R_{client,i}$ only) In RD-LoRA, clients train only their client-specific $R_{client,i}^{(t)}$ while $R_{global}^{(t-1)}$ is fixed. The server aggregates these $R_{client,i}^{(t)}$ to form the next global anchor. This process leads to:

$$R_{global}^{(t)} \xrightarrow{t \to \infty} \frac{1}{1+\gamma} R_{shared}^* \quad (F.A.1)$$

**Scenario B:** Naive Aggregation of Full Local Knowledge (Client trains and uploads the full adapted local matrix) In this baseline, clients directly optimize and upload a full local routing matrix $R_{local,i}^{(t)}$ (approximating $R_i^*$), which the server aggregates. This process leads to:

$$R_{global}^{(t)} \xrightarrow{t \to \infty} R_{shared}^* \quad (F.B.1)$$

Conclusion: Both scenarios converge to the shared knowledge core, but **Scenario A (RD-LoRA) demonstrates significantly stronger and faster convergence due to its inherent contraction mapping and active purification of local knowledge.**

## C.1   PROOF

### C.1.1   SCENARIO A: RD-LoRA (CLIENT TRAINS $R_{client,i}$ AND UPLOADS $R_{client,i}$ ONLY)

(1) Client-side Purification via SCRD:

On the client side, SCRD freezes $R_{global}^{(t-1)}$. **As established in Theorem 3.1, this constraint reduces the norm of the local gradient, compelling the optimizer to find solutions within the semantic subspace defined by the global anchor. Consequently, when the local loss $\mathcal{L}_i$ is minimized, the trainable $R_{client,i}$ necessarily converges to compensate for the global model's deficiencies on local data, yielding a purified local routing matrix in the form of a residual:**

$$R_{client,i}^{(t)} \approx R_i^* - \gamma R_{global}^{(t-1)} \quad (F.A.2)$$

Here, $\gamma$ is a scaling factor ensuring this residual nature.

(2) Server-side Aggregation Dynamics:

During the aggregation phase, we only aggregate the client updates $\{R_{client,i}^{(t)}\}$ from the current round to reconstruct the next global anchor. Thus, $R_{global}^{(t)}$ can be regarded as the shared route distilled from the local knowledge of the current round. The global anchor from the previous round, $R_{global}^{(t-1)}$, solely provides a constraint during the forward pass and is not repeatedly aggregated, thereby preventing the accumulation of historical noise. The server computes the mean:

$$R_{global}^{(t)} \approx \mathbb{E}_i[R_{client,i}^{(t)}] \quad (F.A.3)$$

Substituting (A.2) and $R_i^* = R_{shared}^* + \delta_i$, with $\mathbb{E}_i[\delta_i] = 0$, we get a linear iterative relation:

$$R_{global}^{(t)} \approx R_{shared}^* - \gamma R_{global}^{(t-1)} \quad (F.A.4)$$

(3) Convergence Analysis:

Equation (A.4) is a contraction mapping ($0 < \gamma < 1$). Its fixed point $R_{global}^{(\infty)}$ is $\frac{1}{1+\gamma} R_{shared}^*$. This ensures stable and rapid convergence, where the $-\gamma R_{global}^{(t-1)}$ term actively dampens oscillations and accelerates regularization towards the shared core.

### C.1.2 SCENARIO B: NAIVE AGGREGATION OF FULL LOCAL KNOWLEDGE (CLIENT TRAINS AND UPLOADS THE FULL ADAPTED LOCAL MATRIX)

(1) Local Learning & Aggregation:

Clients optimize and upload a full local matrix $R_{local,i}^{(t)} \approx R_i^*$, without freezing $R_{global}^{(t-1)}$. The server aggregates these: $R_{global}^{(t)} \approx \mathbb{E}_i[R_{local,i}^{(t)}]$ With $R_{local,i}^{(t)} \approx R_i^* = R_{shared}^* + \delta_i$ and $\mathbb{E}_i[\delta_i] = 0$, this leads to:

$$R_{global}^{(t)} \approx R_{shared}^* \quad (F.B.2)$$

(2) Weaker Convergence:

This convergence relies solely on the statistical averaging of zero-mean client-specific biases $\delta_i$. It lacks the active contraction term $(-\gamma R_{global}^{(t-1)})$ of Scenario A, resulting in inherently slower and less stable convergence. Without explicit gradient shielding and residual training, the global state is more susceptible to "knowledge pollution" and "aggregation distortion" from diverse local optima, leading to prolonged oscillations and weaker convergence guarantees compared to RD-LoRA.

**Conclusion:** The local gradient shielding mechanism purifies local knowledge into a pure residual relative to the global model, whilst the server's unbiased aggregation reconstructs the global consensus from these purified residuals in each round. This virtuous cycle, by disentangling local and global knowledge, enables the server to accurately distill shared information from heterogeneous updates, thereby guaranteeing the stable convergence of the global model towards the shared knowledge core.

## D ALGORITHM OF RD-LoRA

---

**Algorithm 1** Alternating Training with SCRD and PCA in RD-LoRA

---

1: **Initialise:** Global LoRA weights $A_{global}^{(0)}$, $B_{global}^{(0)}$, routing anchor $R_{global}^{(0)}$
2: **for** each communication round $t = 0, 1, 2, \ldots$ **do**
3:     **if** $t \bmod 2 = 0$ **then**           ▷ $B$-**round: train** $B$ **and** $R_{client}$
4:         Server broadcasts $\bar{A} = A_{global}^{(t)}$, $R_{global}^{(t)}$ to all clients
5:         **for** each client $i$ **in parallel do**
6:             Freeze $\bar{A}$, train $B_i^{(t)}$, $R_{client,i}^{(t)}$ by minimising:

$$\mathcal{L}_i^{(t)} = \mathbb{E}_{(x,y) \sim \mathcal{D}_i} \left[ \ell\big(f(x; W_0 + B_i^{(t)}(R_{client,i}^{(t)} + \gamma R_{global}^{(t)})\bar{A}), y\big) \right]$$

7:             Upload $B_i^{(t)}$, $R_{client,i}^{(t)}$ to the server
8:         **end for**
9:         Server aggregates $\{B_i^{(t)}\}$ into $B_{global}^{(t+1)}$ via PCA
10:        Server aggregates $\{R_{client,i}^{(t)}\}$ into $R_{global}^{(t+1)}$ via PCA
11:     **else**                 ▷ $A$-**round: train** $A$ **and** $R_{client}$
12:         Server broadcasts $\bar{B} = B_{global}^{(t)}$, $R_{global}^{(t)}$ to all clients
13:         **for** each client $i$ **in parallel do**
14:             Freeze $\bar{B}$, train $A_i^{(t)}$, $R_{client,i}^{(t)}$ by minimising:

$$\mathcal{L}_i^{(t)} = \mathbb{E}_{(x,y) \sim \mathcal{D}_i} \left[ \ell\big(f(x; W_0 + \bar{B}(R_{client,i}^{(t)} + \gamma R_{global}^{(t)})A_i^{(t)}), y\big) \right]$$

15:             Upload $A_i^{(t)}$, $R_{client,i}^{(t)}$ to the server
16:         **end for**
17:         Server aggregates $\{A_i^{(t)}\}$ into $A_{global}^{(t+1)}$ via PCA
18:         Server aggregates $\{R_{client,i}^{(t)}\}$ into $R_{global}^{(t+1)}$ via PCA
19:     **end if**
20: **end for**

---

| Method | Client Computation | Server Computation | Uplink (UL) Comm. | Downlink (DL) Comm. |
|---|---|---|---|---|
| FFA-LoRA | $O(rd^2)$ | $O(Kdr)$ | $dr$ | $dr$ |
| Zero-Padding (FedIT) | $O(rd^2)$ | $O(Kd)$ | $2dr$ | $2dr$ |
| FLoRA | $O(rd^2)$ | $O(Kd^2r)$ | $2dr$ | $d^2$ |
| LoRA-A$^2$ | $O(rd^2)$ | $O(Kdr)$ | $dr$ | $dr$ |
| FlexLoRA | $O(rd^2)$ | $O(Kd^2r + Kd^2 + \boldsymbol{d}^3)$ | $2dr$ | $2dr$ |
| **RD-LoRA (Ours)** | $O(rd^2)$ | $O(Kdr + Kr^2)$ | $dr + r^2$ | $dr + r^2$ |

**Note:** $K$ denotes the number of clients, $d$ represents the input/output dimension, and $r$ is the rank of LoRA, where $r \ll d$. Comm. = communication.

Table 6: Complexity Comparison of Federated LoRA Fine-tuning Methods.

## E    DETAILED COMPARATIVE METHODOLOGY

Zero-padding (Cho et al., 2024) pads lower-rank client LoRAs to the server's maximum rank before aggregation. FLoRA (Wang et al., 2024) stacks heterogeneous LoRAs on the server and aggregates via matrix multiplication. FlexLoRA (Bai et al., 2024) aggregates the BA product and then applies SVD to re-factorise it into modules A and B. LoRA-A$^2$ (Koo et al., 2024) alternately freezes A and B and selects key parameters by client data importance. "Local" denotes the mean performance of clients trained independently on their own data and serves as a baseline to validate FL effectiveness. FFA-LoRA (Sun et al., 2024) assumes identical ranks and updates only the zero-initialised branch while fixing the randomly initialised one; this rank-identity assumption means FFA-LoRA is suitable solely for homogeneous configurations and is inapplicable to heterogeneous-rank federated fine-tuning. Accordingly, we report FFA-LoRA only in Homo comparisons and exclude it from Heter analyses to avoid unfair or misleading comparisons. All baselines are re-implemented on OpenFedLLM (Ye et al., 2024) with matched model, splits, and optimization settings.

## F    COMPLEXITY COMPARISON

We analysed the primary overheads. The analysis of client/server computation and uplink/downlink communication, detailed in the Table 6, demonstrates that our method does not introduce significant costs.

As detailed in the table, we analyse the computational and communication overheads, demonstrating that our method maintains high efficiency without imposing significant costs.

Client Computation: Our method maintains a complexity of $O(rd^2)$, which is consistent with all baseline methods. This ensures that no additional computational burden is placed on resource-constrained client devices. Server Computation: Our server-side complexity is $O(Kdr + Kr^2)$. Given that $r \ll d$, this is on par with highly efficient methods like FFA-LoRA and LoRA-A$^2$. This presents a marked efficiency improvement over both FLoRA and, in particular, FlexLoRA, which incurs a substantial cost of $O(Kd^2r + Kd^2 + \boldsymbol{d}^3)$. By circumventing the need for SVD operations on the full dimension $d$, our method avoids the cubic ($\boldsymbol{d}^3$) complexity inherent to FlexLoRA, thereby reducing server computation by two orders of magnitude and ensuring scalability.

Communication: Both uplink and downlink costs are $dr + r^2$. Given that $r \ll d$, the $r^2$ term is negligible. Consequently, our communication overhead is comparable to the most efficient baselines (FFA-LoRA) and is approximately half the cost of FlexLoRA ($2dr$).

## G    LIMITATIONS AND FUTURE WORK

Cross-rank alignment currently uses zero padding and truncation, causing a small drop when moving to heterogeneous ranks; we will explore adaptive rank matching to mitigate this. The evaluation is limited to English instruction data and two benchmarks, and generalization to larger LLMs and additional LoRA variants remains to be verified. Future work includes adaptive alignment, broader PEFT forms and tasks, and multilingual and alignment evaluations.

## H  USE OF LARGE LANGUAGE MODELS

Large Language Models (LLMs), such as ChatGPT, were employed to assist with the translation and linguistic refinement of certain sections of the paper, helping ensure clarity, coherence, and consistency in the use of academic English throughout. The tool did not generate research ideas, methods, analyses, results, or conclusions, and it did not alter the scientific content. All technical content and analysis presented in this paper are the sole work of the authors.

