# OpenReview forum: "Routing-Deconstructed LoRA in Federated Fine-Tuning"
_ICLR.cc/2026/Conference — Submitted to ICLR 2026_

### Official Review · Reviewer_wjiw · 2025-10-30

**Soundness:** 3
**Presentation:** 3
**Contribution:** 2
**Rating:** 4
**Confidence:** 5

**Summary:**

The paper proposes RD-LoRA for federated LoRA fine-tuning under client/data heterogeneity. It builds on alternating freezing (only A or B is updated per round) to lower communication and aggregation noise, and introduces a routing matrix split to decouple shared semantics from client-specific signals and reduce knowledge contamination. On the server, a PCA applies position-wise adaptive weights with historical regularization to mitigate aggregation distortion when fusing A/B/R. The method handles heterogeneous ranks via zero-padding/truncation before aggregation and redistribution. Experiments on Llama-2 and TinyLlama report consistent gains over baselines in both homogeneous and heterogeneous settings, with communication similar to alternating-freeze methods.

**Strengths:**

The paper clearly pinpoints three key challenges in federated LoRA fine-tuning: aggregation noise, knowledge contamination, and aggregation distortion. The solution proposed focus on the specific challenges and has clear motivation.

Empirical results are strong and consistent across Llama-2 and TinyLlama in both homogeneous and heterogeneous rank settings, with low communication comparable to alternating-freeze methods and ablations showing that removing SCRD or PCA hurts performance.

The paper clearly states the strength and weakness of previous works, exploring the accuracy, communication, and heterogeneity.

**Weaknesses:**

The pipeline is overly complex for practical LLM fine-tuning, hinging on alternating A/B rounds, an extra routing matrix, and server-side PCA; moreover, the experiments use very heavy schedules (e.g., 200 communication rounds with 10 local epochs per round), which may not be necessary in many real deployments and can confound the contribution of the routing/PCA design.

The treatment of heterogeneous ranks lacks a stronger theoretical underpinning: cross-rank alignment ultimately relies on zero-padding or truncation, which the paper itself acknowledges as a limitation and potential source of degradation, this seems partly at odds with the broader claim of mitigating aggregation noise.

Some figures and expressions are not sufficiently precise; for instance, the RD-LoRA overview (Figure 2) is visually crowded with many modules/arrows, making the flow difficult to understand quickly.

**Questions:**

Could you report the test accuracy trajectory over all 200 communication rounds, including variance across seeds, and clarify at which round RD-LoRA first surpasses baselines?

What is the base model’s zero-shot (or supervised) accuracy on each benchmark before any federated fine-tuning, so we can quantify absolute gains?

How does the method handle client selection or partial participation per round (e.g., random subset of clients)? Please specify any changes to routing updates, PCA aggregation, and convergence behavior under varying participation rates.

---

> ### Author Response · Authors · 2025-11-24
> **Point-to-point response to wjiw (1/4)**
>
> We thank reviewer wjiw for the valuable time and constructive feedback. We provide point-to-point response below.
>
> >**Q1: The pipeline is overly complex for practical LLM fine-tuning, hinging on alternating A/B rounds, an extra routing matrix, and server-side PCA; moreover, the experiments use very heavy schedules (e.g., 200 communication rounds with 10 local epochs per round), which may not be necessary in many real deployments and can confound the contribution of the routing/PCA design. / Could you report the test accuracy trajectory over all 200 communication rounds, including variance across seeds, and clarify at which round RD-LoRA first surpasses baselines?**
>
> **A1:** We thank the reviewer for this valuable critique. We clarify that the A/B alternating rounds are an established baseline technique, adopted from Koo et al[d]. Our core innovations are two orthogonal modules built upon this foundation: client-side SCRD, which utilizes a routing matrix to manage knowledge, and server-side PCA Aggregation.
>
> Following your suggestion, we have included  complete 200-round test accuracy curves for RD-LoRA and all baselines on the Dolly dataset, which we have added as **Figure 6** in the revision. Each curve represents the mean of five runs with different random seeds, with shaded areas indicating standard deviation.
>
> The results clearly demonstrate that RD-LoRA achieves superior convergence, consistently surpassing the strongest baseline, FlexLoRA, from round 25 onwards with a growing advantage.
>
> **References**
> - [d] Koo J, Jang M, Ok J. Towards robust and efficient federated low-rank adaptation with heterogeneous clients[C]//Proceedings of the 63rd Annual Meeting of the Association for Computational Linguistics (Volume 1: Long Papers). 2025: 416-429.

---

> ### Author Response · Authors · 2025-11-24
> **Point-to-point response to wjiw (2/4)**
>
> >**Q2: The treatment of heterogeneous ranks lacks a stronger theoretical underpinning: cross-rank alignment ultimately relies on zero-padding or truncation, which the paper itself acknowledges as a limitation and potential source of degradation, this seems partly at odds with the broader claim of mitigating aggregation noise.**
>
> **A2:** We thank the reviewer for this critical observation, which allows us to clarify a key theoretical point. We contend that zero-padding, while a practical necessity, does not introduce a new, additive source of aggregation noise. Instead, it serves as a technical pre-requisite to enable the computation of the cross-term bias across heterogeneous ranks. The fundamental structure of this noise term remains identical to the homogeneous case.
>
> + **1. Formal Derivation and Comparison:**
> Let's first revisit the aggregation noise in the **homogeneous setting** (all ranks equal to $r$), as defined:
> \begin{aligned}
> \Delta W_{\text{FedIT}} = \left(\frac{1}{k}\sum_{i=1}^{k} B_i\right) \left(\frac{1}{k}\sum_{i=1}^{k} A_i\right) \
> = \frac{1}{k^2}\sum_{i=1}^{k} (B_i A_i) + \underbrace{\frac{1}{k^2}\sum_{i \neq j}^{k} (B_i A_j)}_{\text{aggregation noise}} \quad(3)
> \end{aligned}
>
> The cross-terms between the LoRA modules from different clients give rise to aggregation noise. Furthermore, by applying the $\frac{1}{k}$ scaling factor to both $B_i$ and $A_i$ independently, FedIT introduces an erroneous $\frac{1}{k}$ coefficient to the ideal update component, thereby exacerbating the error in the LoRA aggregation.
>
>  Now, consider the **heterogeneous setting**. Let client $i$ have rank $r_i$. We pad all matrices to a common maximal rank $R = \max_k(r_k)$. The padded matrices, denoted $B^\prime_i$ and $A^\prime_i$, have the following block matrix structure:
>
> $$
> B^\prime_i = [ B_i | \boldsymbol{0}_{d \times (R-r_i)} ] \in R^{d \times R}
> $$
> and
>
> $$
> A^\prime_i =[\frac{A_i}{\boldsymbol{0}_{(R-r_i) \times l}}] \in R^{R \times l}
> $$
>
> The aggregated update, $\Delta W_{\text{Hetero}}$, is formed by averaging these padded matrices:
> $$
> \Delta W_{\text{Hetero}} = \left(\frac{1}{k}\sum_{i=1}^{k} B^\prime_i\right) \left(\frac{1}{k}\sum_{j=1}^{k} A^\prime_j\right)
> $$
> Expanding this product gives:
>
> $$
> \Delta W_{Hetero} = \frac{1}{k^2} \sum_{i=1}^{k} \sum_{j=1}^{k} B^\prime_i A^\prime_j
> $$
>
> $$
> = \frac{1}{k^2}\sum_{i=1}^{k} (B^\prime_i A^\prime_i) + \frac{1}{k^2} \sum_{i \neq j}^{k} B^\prime_i A^\prime_j
> $$
>
>
>
> + **2. Crucial Analysis:**
> The signal term is perfectly preserved. The zero-padding does not alter the original update matrix:
>     $$
>     B^\prime_iA^\prime_j= \left[ B_i \mid \mathbf{0} \right] \begin{bmatrix} A_i \\ \mathbf{0} \end{bmatrix} = B_i A_i + \mathbf{0} = B_i A_i.
>     $$
> The aggregation noise term, $\frac{1}{k^2} \sum_{i \neq j}^{k} B^\prime_iA^\prime_j$, arises from the cross-multiplication of matrices from different clients. This is fundamentally the same source of error as in the homogeneous case. Zero-padding is merely the algebraic mechanism that allows the product $B'_i A'_j$ to be computed in the common $R$-dimensional space.
>
> + **3. Conclusion:** Zero-padding does not introduce a new, independent error term; it simply defines the algebra for computing the existing cross-term bias when ranks differ. Therefore, our claim of mitigating aggregation noise remains fully consistent, as our core methods (alternating updates and PCA) are designed to combat this fundamental cross-term bias, regardless of whether the ranks are homogeneous or made homogeneous via padding. The alternating freeze iteration method we use in this paper can avoid this noise.

---

> ### Author Response · Authors · 2025-11-24
> **Point-to-point response to wjiw (3/4)**
>
> >**Q3: Some figures and expressions are not sufficiently precise; for instance, the RD-LoRA overview (Figure 2) is visually crowded with many modules/arrows, making the flow difficult to understand quickly.**
>
> **A3:** We thank the reviewer for this valuable feedback on the clarity of our visualisations. We fully agree that an intuitive and clear illustration is paramount for the rapid comprehension of a complex method, and we apologise that the original Figure 2 did not meet this standard. Taking this feedback to heart, we will therefore completely redesign Figure 2 in the revised manuscript. We thank the reviewer again for this constructive suggestion.
>
> ---
>
> >**Q4: What is the base model’s zero-shot (or supervised) accuracy on each benchmark before any federated fine-tuning, so we can quantify absolute gains?**
>
> **A4:** We thank the reviewer for this excellent suggestion. To clearly quantify the absolute performance gains, we have now included the zero-shot accuracy of each base model in our experimental tables. The analysis confirms that RD-LoRA consistently delivers significant improvements over strong, untuned base models across various scales and tasks.
>   **Table D4.1: Performance with Multiple Seeds (Homogeneous Setting, Llama2-7B)**
> | Method | Wizard (MMLU) | Dolly (MMLU) | Alpaca (MMLU) |
> | :--- | :---: | :---: | :---: |
> | Zero-shot | 35.09 | 35.09 | 35.09|
> | FFA-LoRA | 34.68 | 44.32 | 32.58 |
> | Zero-padding (FedIT) | 34.73 | 44.11 | 32.87 |
> | FLoRA | 35.78 | 45.27 | 35.11 |
> | LoRA-$\mathrm{A}^2$ | 35.81 | 45.69 | 35.71 |
> | FlexLoRA | 36.07 | 45.72 | 35.83 |
> | **RD-LoRA (Ours)** | **36.68** | **47.70** | **37.58** |
>
> **Table D4.2: Performance with Multiple Seeds (Heterogeneous Setting, Llama2-7B)**
> | Method | Wizard (MMLU) | Dolly (MMLU) | Alpaca (MMLU) |
> | :--- | :---: | :---: | :---: |
> | Zero-shot | 35.09 | 35.09 | 35.09 | 3.73 |
> | FFA-LoRA | 34.33 | 43.55 | 31.74 |
> | Zero-padding (FedIT) | 34.47 | 43.77 | 32.26 |
> | FLoRA | 35.44 | 44.94 | 34.57 |
> | LoRA-$\mathrm{A}^2$ | 35.64 | 45.38 | 35.62 |
> | FlexLoRA | 36.17 | 45.91 | 36.20 |
> | **RD-LoRA (Ours)** | **36.54** | **47.16** | **37.21** |
>
> **Table D4.3: Performance Comparison on Qwen3-8B-Base (Heterogeneous Setting)**
> | Method (Heter) |  Dolly (MMLU) | BigMath (AIME 25) |
> | :--- |  :---: | :---: |
> | **Zero-shot Base** |  72.49 | 17.11 |
> | Zero-padding (FedIT)  | 71.09 |   16.07   |
> | FlexLoRA | 73.14 |    18.46   |
> | **RD-LoRA (Ours)** | **74.11** | **19.73** |
>
> **Table D4.4: Performance Comparison on Llama4-Scout-17B (Heterogeneous Setting)**
> | Method (Heter) | Dolly (MMLU) | BigMath (AIME 25) |
> | :--- | :---: | :---: |
> | **Zero-shot Base** | 75.13 | 9.02 |
> | Zero-padding (FedIT)  | 73.64 |   8.85   |
> | FlexLoRA | 75.67 |  11.03   |
> | **RD-LoRA (Ours)** | **77.10** | **12.21** |
>
>
> *   **On Llama2-7B (Tables 1 & 2):** The base model achieves a zero-shot MMLU score of 35.09. Our RD-LoRA method demonstrates substantial gains, boosting performance by up to **+12.61%** on the Dolly dataset. Notably, this analysis also reveals that simpler federated methods can sometimes degrade performance below this baseline, highlighting the challenges our method successfully overcomes.
>
> *   **On Larger Models (Tables 3 & 4):** This trend holds for more capable models. On Qwen3-8B, RD-LoRA improves the strong MMLU baseline from 72.49 to 74.11 and boosts the challenging BigMath score by **+2.62%**. Similarly, for the Llama4-Scout-17B model, our method provides  a remarkable **+3.19%** improvement on BigMath over its already high baseline.
>
> These results validate that RD-LoRA not only outperforms other federated methods but also consistently adds significant value over the powerful zero-shot capabilities of the base models.

---

> ### Author Response · Authors · 2025-11-24
> **Point-to-point response to wjiw (4/4)**
>
> >**Q5: How does the method handle client selection or partial participation per round (e.g., random subset of clients)? Please specify any changes to routing updates, PCA aggregation, and convergence behavior under varying participation rates.**
>
> **A5:** We thank the reviewer for this crucial question regarding the practicalities of partial client participation. We shall address this question by examining three key aspects: RD-LoRA to client selection, Component interaction with participation rate, and client scaling experiments.
>
> + **1. RD-LoRA to Client Selection:**
> We employ a standard partial participation protocol, **randomly sampling** two of ten clients per round. This ratio is applied consistently across all methods for a fair comparison. Within each round, all updates and aggregation are performed exclusively on this active subset.
>
> + **2. Component Interaction with Participation Rate:**
> Our method is inherently robust to varying participation rates and does not rely on a specific ratio:
>   + **a. Client-side SCRD:** Each participating client's update ($R_{client,i}^{(t)}$) is purified as a residual against the fixed $R_{global}^{(t-1)}$, making local contributions robust regardless of the active subset size. This mechanism helps alleviate the impact of single update biases on the global model.
>   + **b. Server-side PCA Aggregation:** More participating clients offer richer information, allowing PCA to more robustly estimate $R_{global}$ and potentially accelerate convergence. Conversely, even with fewer clients or dropouts, PCA's denoising capabilities effectively extract salient shared knowledge from the available updates, mitigating aggregation distortion.
>
> + **3. Client Scaling Experiments:**
> We conducted new experiments with expanded client pools. We evaluated two scenarios: a pool of 10 total clients (sampling 2, 4, and 8 per round) and a larger pool of 50 total clients (sampling 20 and 40 per round). The results, presented in the table below.
>
>
>   **Table D5.1: MMLU Performance on Dolly with an Expanded Client Pool (Llama2, Heterogeneous)**
>  | Method             | Sampled 2 (Total 10) | Sampled 4 (Total 10) | Sampled 8 (Total 10) | Sampled 20 (Total 50) | Sampled 40 (Total 50) |
>  | :----------------- | :------------------: | :------------------: | :------------------: | :-------------------: | :-------------------: |
>  | FlexLoRA           |        45.91         |        47.46         |        48.83         |         46.11         |         47.57         |
>  | **RD-LoRA (ours)** |   **47.16 (+1.25)**  |   **49.14 (+1.68)**  |   **50.21 (+1.38)**  |   **47.87 (+1.76)**   |   **49.30 (+1.73)**   |
>
>   The analysis confirms RD-LoRA's consistent superiority over FlexLoRA. Our method achieves both the peak accuracy of 50.21% (in the 10-client/8-participant setting) and the largest performance gap of 1.76% (in the 50-client/20-participant setting). This demonstrates that our performance advantage is maintained or even increased as more clients are sampled per round.
>
> + **4. Conclusion:** Lower participation rates inherently increase sampling bias, a challenge for which RD-LoRA's design offers superior robustness. Unlike FedAvg, our Poly-Consensus Aggregation (PCA) is less susceptible to this noise. Specifically, its Historical Steady Alignment regulariser prevents drastic model oscillations, while its Omni-Scale Integration Module selectively amplifies the most reliable information within the participating subset.
>
> We appreciate again your thoughtful review and we hope we addressed your concerns. Please let us know if you'd like any further information.

---

### Official Review · Reviewer_8fzm · 2025-10-31

**Soundness:** 3
**Presentation:** 3
**Contribution:** 2
**Rating:** 4
**Confidence:** 4

**Summary:**

This paper proposes Routing-Deconstructed LoRA (RD-LoRA), a federated fine-tuning framework for large language models that introduces two main components: the Server–Client Routing Deconstructor (SCRD) to separate global and local knowledge, and the Poly-Consensus Aggregation (PCA) mechanism, which employs adaptive weighted averaging for aggregation under data heterogeneity. The work aims to alleviate aggregation noise, knowledge contamination, and aggregation distortion in federated LoRA fine-tuning.

**Strengths:**

- The proposed method demonstrates clear empirical effectiveness, especially under non-IID settings, and achieves superior performance over several strong baselines.
- The experimental section is comprehensive in terms of datasets, models, and metrics, providing a convincing empirical comparison.

**Weaknesses:**

- Equation (2) is imprecise and potentially misleading. $\frac{1}{k}\sum \Delta W_i$ is not directly equal to $\frac{1}{k}\sum \Delta B_i \times \frac{1}{k}\sum \Delta A_i$ but somehow approximated by it.
- The discussion of bias from averaging B and A (Equation 3) lacks depth. The paper shows that the left-hand side differs from the right-hand side but does not analyze whether this necessarily constitutes a biased or sub-optimal estimate. More intuition or references to prior analyses of LoRA aggregation bias would strengthen the argument.
- The reasoning of theorem 3.1 of “ effectiveness of SCRD arising from reduced gradient scale” is unclear. The theoretical analysis focuses solely on local updates, while in federated settings $R_{\text{global}}$ is produced through server-side aggregation. It is not evident how the local analysis guarantees separation between local and global knowledge in practice.
- A key baseline, FedIT (FedAvg + LoRA), is missing. Given that RD-LoRA claims novelty on aggregation scheme, direct comparison with FedIT is essential as a comparison with the most naive method is necessary.
- Only two clients out of ten are sampled per communication round, which may not adequately represent real-world FL participation patterns and can bias results.
- No indication is given that experiments were repeated with multiple random seeds. Reporting variance would improve reliability.

**Questions:**

- **OSIM implementation:** The paper does not describe how the per-branch logits $s_i[u,v]$ in the Omni-Scale Integration Module are computed.
- **PCA specificity:** The proposed Poly-Consensus Aggregation seems generally applicable to any matrix aggregation, not specifically tailored to LoRA. Can author share more insights on how LoRA can be beneficial on PCA?
- **Assumption in Theorem 3.1:** Is it realistic to assume the symmetric part of $R_{\text{global}}^\top R_{\text{client}}$ is positive definite?

---

> ### Author Response · Authors · 2025-11-24
> **Point-to-point response to 8fzm (1/6)**
>
> We thank reviewer 8fzm for the valuable time and constructive feedback. We provide point-to-point response below.
>
>
> >**Q1: Equation (2) is imprecise and potentially misleading. $\frac{1}{k}\sum \Delta W_i$  is not directly equal to $\frac{1}{k}\sum \Delta B_i \times \frac{1}{k}\sum \Delta A_i$ but somehow approximated by it.**
>
> **A1:** We sincerely thank the reviewer for this precise and insightful feedback. Your valuable suggestions have significantly enhanced the clarity and overall quality of our manuscript. Our intention with Equation (2) is to make clear the gap between the ideal global FedAvg update on LoRA and the practical implementation in FedIT methods[c]. To sharpen this, we now explicitly define two terms: $\Delta W_{\text{Ideal}}$ for the ideal aggregation of the full updates, and $\Delta W_{\text{FedIT}}$ for the naive averaging of the low-rank factors separately. Our revised Equation (2) now correctly and unambiguously states the non-equivalence between these two approaches:
> $$
> \Delta W_{\text{FedIT}} =  \left(\frac{1}{k}\sum_{i=1}^{k} B_i\right) \left(\frac{1}{k}\sum_{i=1}^{k} A_i\right) \neq \Delta W_{\text{Ideal}} =  \frac{1}{k}\sum_{i=1}^{k} (B_i A_i) \quad (2)
> $$
>   This corrected formulation provides a more rigorous foundation for our analysis of aggregation noise. We are grateful for your guidance in making this crucial improvement.
>
> ---
>
> >**Q2: The discussion of bias from averaging B and A (Equation 3) lacks depth. The paper shows that the left-hand side differs from the right-hand side but does not analyze whether this necessarily constitutes a biased or sub-optimal estimate. More intuition or references to prior analyses of LoRA aggregation bias would strengthen the argument.**
>
> **A2:** We thank the reviewer for highlighting the need for a deeper analysis of aggregation bias. In response, we have thoroughly revised this section to provide a more rigorous argument. Specifically, we have expanded the expression for $\Delta W_{\text{FedIT}}$ in the manuscript
>
> \begin{aligned}
> \Delta W_{\text{FedIT}} = \left(\frac{1}{k}\sum_{i=1}^{k} B_i\right) \left(\frac{1}{k}\sum_{i=1}^{k} A_i\right) \
> = \frac{1}{k^2}\sum_{i=1}^{k} (B_i A_i) + \underbrace{\frac{1}{k^2}\sum_{i \neq j}^{k} (B_i A_j)}_{\text{aggregation noise}} \quad(3)
> \end{aligned}
>
>   The cross-terms between the LoRA modules from different clients give rise to aggregation noise. Furthermore, by applying the $\frac{1}{k}$ scaling factor to both $B_i$ and $A_i$ independently, FedIT introduces an erroneous $\frac{1}{k}$ coefficient to the ideal update component, thereby exacerbating the error in the LoRA aggregation.
>
> **References:**
> - [c] Sun Y, Li Z, Li Y, et al. Improving LoRA in Privacy-preserving Federated Learning[C]//The Twelfth International Conference on Learning Representations.

---

> > ### Author Response · Authors · 2025-11-24
> > **Point-to-point response to 8fzm (2/6)**
> >
> > >**Q3: The reasoning of theorem 3.1 of “ effectiveness of SCRD arising from reduced gradient scale” is unclear. The theoretical analysis focuses solely on local updates, while in federated settings R_global is produced through server-side aggregation. It is not evident how the local analysis guarantees separation between local and global knowledge in practice.**
> >
> > **A3:** We thank the reviewer for this profound and critical question. To address the connection between the local analysis in Theorem 3.1 and the global guarantee of knowledge separation, we now provide a formal analysis in the newly introduced Theorem 3.2. We have incorporated this theorem into the revised manuscript.
> >
> > **Theorem 3.2 (Comparative Analysis of Global Routing Matrix Convergence)**
> >   + **Assume** the ideal optimal routing matrix  $R_i^\*$   for each client $i$ decomposes into a **shared knowledge core** $R_{shared}^\*$ and a zero-mean, **client-specific bias** $\delta_i$ (i.e., $R_i^\* = R_{shared}^\* + \delta_i$, with $E_i[\delta_i] = 0$).
> >   We compare two scenarios for updating the global routing anchor $R_{global}^{(t)}$:
> >
> > + **Scenario A: RD-LoRA (Client trains $R_{client,i}$ and uploads $R_{client,i}$ only)**
> > In RD-LoRA, clients train only their client-specific $R_{client,i}^{(t)}$ while $R_{global}^{(t-1)}$ is fixed. The server aggregates these $R_{client,i}^{(t)}$ to form the next global anchor. This process leads to:
> > $$
> > R_{global}^{(t)} \xrightarrow{t \to \infty} \frac{1}{1+\gamma} R_{shared}^\* \quad (A.1)
> > $$
> >
> > + **Scenario B: Naive Aggregation of Full Local Knowledge (Client trains and uploads the full adapted local matrix)**
> > In this baseline, clients directly optimize and upload a full local routing matrix $R_{local,i}^{(t)}$ (approximating $R_i^*$), which the server aggregates. This process leads to:
> > $$
> > R_{global}^{(t)} \xrightarrow{t \to \infty} R_{shared}^\* \quad (B.1)
> > $$
> >
> > + **Conclusion:** Both scenarios converge to the shared knowledge core, but **Scenario A (RD-LoRA) demonstrates significantly stronger and faster convergence due to its inherent contraction mapping and active purification of local knowledge.**
> >
> >
> >
> > **Proof**
> >
> > + **Scenario A: RD-LoRA (Client trains $R_{client,i}$ and uploads $R_{client,i}$ only)**
> >
> >   + **Client-side Purification via SCRD:** On the client side, SCRD freezes $R_{global}^{(t-1)}$. As established in Theorem 3.1, this constraint reduces the norm of the local gradient, compelling the optimizer to find solutions within the semantic subspace defined by the global anchor. Consequently, when the local loss $L_i$ is minimized, the trainable $R_{client,i}$ necessarily converges to compensate for the global model's deficiencies on local data, yielding a purified local routing matrix in the form of a residual:
> > $$R_{client,i}^{(t)} \approx R_i^\* - \gamma R_{global}^{(t-1)} \quad (A.2)$$
> >     Here, $\gamma$ is a scaling factor ensuring this residual nature.
> >
> >   + **Server-side Aggregation Dynamics:** During the aggregation phase, we only aggregate the client updates $\{R_{client,i}^{(t)}\}$ from the current round to reconstruct the next global anchor. Thus, $R_{global}^{(t)}$ can be regarded as the shared route distilled from the local knowledge of the current round. The global anchor from the previous round, $R_{global}^{(t-1)}$, solely provides a constraint during the forward pass and is not repeatedly aggregated, thereby preventing the accumulation of historical noise. The server computes the mean:
> > $$
> > R_{global}^{(t)} \approx E_i[R_{client,i}^{(t)}] \quad (A.3)
> >     $$
> >     Substituting (A.2) and $R_i^\* = R_{shared}^\* + \delta_i$, with $E_i[\delta_i] = 0$, we get a linear iterative relation:
> > $$
> > R_{global}^{(t)} \approx R_{shared}^\* - \gamma R_{global}^{(t-1)} \quad (A.4)
> > $$
> >
> >    + **Convergence Analysis:** Equation (A.4) is a **contraction mapping** ($0 < \gamma < 1$). Its fixed point $R_{global}^{(\infty)}$ is $\frac{1}{1+\gamma} R_{shared}^*$. This ensures stable and rapid convergence, where the $-\gamma R_{global}^{(t-1)}$ term actively dampens oscillations and accelerates regularization towards the shared core.

---

> ### Author Response · Authors · 2025-11-24
> **Point-to-point response to 8fzm (3/6)**
>
> + **Scenario B: Naive Aggregation of Full Local Knowledge (Client trains and uploads the full adapted local matrix)**
>
>   + **Local Learning & Aggregation:** Clients optimize and upload a full local matrix $R_{local,i}^{(t)} \approx R_i^\*$, without freezing $R_{global}^{(t-1)}$. The server aggregates these: $R_{global}^{(t)} \approx E_i[R_{local,i}^{(t)}]$ with $R_{local,i}^{(t)} \approx R_i^\* = R_{shared}^* + \delta_i$ and $E_i[\delta_i] = 0$, this leads to:
>     $$
>     R_{global}^{(t)} \approx R_{shared}^\* \quad (B.2)
>     $$
>
>   + **Weaker Convergence:** This convergence relies *solely* on the statistical averaging of zero-mean client-specific biases $\delta_i$. It lacks the active contraction term ($-\gamma R_{global}^{(t-1)}$) of Scenario A, resulting in inherently slower and less stable convergence. Without explicit gradient shielding and residual training, the global state is more susceptible to "knowledge pollution" and "aggregation distortion" from diverse local optima, leading to prolonged oscillations and weaker convergence guarantees compared to RD-LoRA.
>
>   **Conclusion**: The local gradient shielding mechanism purifies local knowledge into a pure residual relative to the global model, whilst the server's unbiased aggregation reconstructs the global consensus from these purified residuals in each round. This virtuous cycle, by disentangling local and global knowledge, enables the server to accurately distil shared information from heterogeneous updates, thereby guaranteeing the stable convergence of the global model towards the shared knowledge core.
>
> ---
>
> >**Q4: A key baseline, FedIT (FedAvg + LoRA), is missing. Given that RD-LoRA claims novelty on aggregation scheme, direct comparison with FedIT is essential as a comparison with the most naive method is necessary.**
>
> **A4:** We sincerely thank the reviewer for raising this crucial point regarding the FedIT baseline.We would like to clarify that the FedIT (FedAvg + LoRA) baseline is indeed included and thoroughly evaluated in our experiments, where it is currently named “Zero-padding”. To enhance clarity in the revised manuscript, we will rename this baseline to Zero-padding (FedIT).
>
> + **Justification for Renaming:**
>   - In the homogeneous rank setting, where all clients share an identical LoRA rank, there is no need to align matrix dimensions. As a result, the aggregation process for the 'Zero-padding' baseline degenerates to the pure FedIT algorithm.
>   - In the more challenging heterogeneous rank setting, FedIT, as a conceptual framework, must first address the issue of mismatched LoRA matrix dimensions before aggregation can be performed. The most direct and straightforward solution is precisely to use zero-padding to align all low-rank matrices to the maximum rank, followed by federated averaging.
>
>   **Table C4.1: Performance with Multiple Seeds (Homogeneous Setting, Llama2-7B)**
>     | Method | Wizard (MMLU) | Dolly (MMLU) | Alpaca (MMLU) |  Alpaca (MT-Bench) |
>     | :--- | :---: | :---: | :---: |  :---: |
>     | Zero-padding (FedIT) | 34.73±0.32 | 44.11±0.43 | 32.87±0.20 | 3.622±0.036 |
>     | **RD-LoRA (Ours)** | **36.68±0.13** | **47.70±0.31** | **37.58±0.20** | **4.224±0.021** |
>
>   **Table C4.2: Performance with Multiple Seeds (Heterogeneous Setting, Llama2-7B)**
>     | Method | Wizard (MMLU) | Dolly (MMLU) | Alpaca (MMLU) | Alpaca (MT-Bench) |
>     | :--- | :---: | :---: | :---: | :---: |
>     | Zero-padding (FedIT) | 34.47±0.40 | 43.77±0.52 | 32.26±0.36 | 3.569±0.034 |
>     | **RD-LoRA (Ours)** | **36.54±0.17** | **47.16±0.26** | **37.21±0.22** |**4.205±0.024** |

---

> ### Author Response · Authors · 2025-11-24
> **Point-to-point response to 8fzm (4/6)**
>
> **A5:** We thank the reviewer for this important question on scalability. To address this, we conducted new experiments with expanded client pools. We evaluated two scenarios: a pool of 10 total clients (sampling 2, 4, and 8 per round) and a larger pool of 50 total clients (sampling 20 and 40 per round). The results, presented in the table below.
>
>   **Table C5.1: MMLU Performance on Dolly with an Expanded Client Pool (Llama2, Heterogeneous)**
>   | Method             | Sampled 2 (Total 10) | Sampled 4 (Total 10) | Sampled 8 (Total 10) | Sampled 20 (Total 50) | Sampled 40 (Total 50) |
> | :----------------- | :------------------: | :------------------: | :------------------: | :-------------------: | :-------------------: |
> | FlexLoRA           |        45.91         |        47.46         |        48.83         |         46.11         |         47.57         |
> | **RD-LoRA (ours)** |   **47.16 (+1.25)**  |   **49.14 (+1.68)**  |   **50.21 (+1.38)**  |   **47.87 (+1.76)**   |   **49.30 (+1.73)**   |
>
> The analysis confirms RD-LoRA's consistent superiority over FlexLoRA. Our method achieves both the peak accuracy of 50.21 (in the 10-client/8-participant setting) and the largest performance gap of 1.76 points (in the 50-client/20-participant setting). This demonstrates that our performance advantage is maintained or even increased as more clients are sampled per round. Our method is primarily designed for cross-silo scenarios, such as collaborations between hospitals, for which a client pool of around 50 would generally be sufficient.
>
> ---
>
> >**Q6: No indication is given that experiments were repeated with multiple random seeds. Reporting variance would improve reliability.**
>
> **A6:** We thank the reviewer for this crucial point. To rigorously assess the stability of our findings, we have now re-run our core Llama2-7B experiments five times, each with a different random seed, to correct for this omission.
>
>   **Table C6.1: Performance with Multiple Seeds (Homogeneous Setting, Llama2-7B)**
>
> | Method | Wizard (MMLU) | Dolly (MMLU) | Alpaca (MMLU) | Alpaca (MT-Bench) |
> | :--- | :---: | :---: | :---: | :---: |
> | Zero-shot | 35.09±0.00 | 35.09±0.00 | 35.09±0.00|  3.730±0.000 |
> | Local | 33.11±0.33 | 40.02±0.53 | 32.10±0.23 |  2.934±0.032 |
> | FFA-LoRA | 34.68±0.27 | 44.32±0.51 | 32.58±0.11 | 3.606±0.037  |
> | Zero-padding (FedIT) | 34.73±0.32 | 44.11±0.43 | 32.87±0.20 | 3.622±0.036|
> | FLoRA | 35.78±0.26 | 45.27±0.57 | 35.11±0.13 | 3.874±0.026  |
> | LoRA-$\mathrm{A}^2$ | 35.81±0.31 | 45.69±0.23 | 35.71±0.24 |  4.043±0.031 |
> | FlexLoRA | 36.07±0.18 | 45.72±0.34 | 35.83±0.18 |  4.102±0.025 |
> | **RD-LoRA (Ours)** | **36.68±0.13** | **47.70±0.31** | **37.58±0.20** |  **4.224±0.021** |
>
>  **Table C6.2: Performance with Multiple Seeds (Heterogeneous Setting, Llama2-7B)**
>
> | Method | Wizard (MMLU) | Dolly (MMLU) | Alpaca (MMLU) |  Alpaca (MT-Bench) |
> | :--- | :---: | :---: | :---: | :---: |
> | Zero-shot | 35.09±0.00 | 35.09±0.00 | 35.09±0.00 |3.730±0.000 |
> | Local | 32.63±0.38 | 39.11±0.57 | 31.94±0.28 |  2.851±0.030 |
> | FFA-LoRA | 34.33±0.31 | 43.55±0.56 | 31.74±0.27 |  3.584±0.038 |
> | Zero-padding (FedIT) | 34.47±0.40 | 43.77±0.52 | 32.26±0.36 |  3.569±0.034 |
> | FLoRA | 35.44±0.34 | 44.94±0.47 | 34.57±0.22 | 3.801±0.034 |
> | LoRA-$\mathrm{A}^2$ | 35.64±0.33 | 45.38±0.27 | 35.62±0.31 |  4.026±0.035 |
> | FlexLoRA | 36.17±0.19 | 45.91±0.38 | 36.20±0.24 |  4.131±0.026 |
> | **RD-LoRA (Ours)** | **36.54±0.17** | **47.16±0.26** | **37.21±0.22** |  **4.205±0.024** |
>
> The results across five seeds confirm RD-LoRA's superior performance and robust stability; for instance, on the Dolly dataset (homogeneous), our method's score of 47.70 surpasses the strongest baseline by nearly two full points while also demonstrating the lowest variance (±0.13) on the Wizard dataset. These comprehensive results will be incorporated into the main paper to validate our findings.

---

> ### Author Response · Authors · 2025-11-24
> **Point-to-point response to 8fzm (5/6)**
>
> >**Q7: OSIM implementation: The paper does not describe how the per-branch logits $s_i[u,v]$ in the Omni-Scale Integration Module are computed. (Question 1)**
>
> **A7:** We thank the reviewer for this insightful question. We acknowledge the omission of implementation details for the Omni-Scale Integration Module (OSIM), which impacts reproducibility. We will add the formal equations and a complete description to the revised manuscript.
>
> In our implementation, OSIM is a lightweight, parameter-shared scoring network. To compute the logit, OSIM first constructs a 3-dimensional feature vector $z_i[u,v]$ for each client $i$ at each position $[u,v]$. This vector is formed by concatenating three key scalar values: the local value $F_i[u,v]$, the global context $F_0[u,v]$, and their deviation $F_i[u,v] - F_0[u,v]$. Its formal definition is:
>
> $$
> z_i[u,v] = \text{concat}(F_i[u,v], F_0[u,v], F_i[u,v] - F_0[u,v]) \in \mathbb{R}^3.
> $$
>
> This feature vector is then processed by a small two-layer Multi-Layer Perceptron (MLP), whose parameters $\phi = \{\mathcal{W}_1, \beta_1, \mathcal{W}_2, \beta_2\}$ are fully shared across all positions, clients, and even LoRA sub-modules (A/B/R). The computation is as follows:
>
> $$
> h_i[u,v] = \text{ReLU}(\mathcal{W}_1 z_i[u,v] + \beta_1) , \qquad s_i[u,v] = \mathcal{W}_2^T h_i[u,v] + \beta_2 \in \mathbb{R}^3.
> $$
>
> The resulting logit $s_i[u,v]$ is then fed into the softmax function in $w_i[u,v] = \frac{\exp\!\big(s_i[u,v]/\tau\big)}{\sum_{j\in I_t^{\theta}}\exp\!\big(s_j[u,v]/\tau\big)}$ of our paper to produce the final aggregation weights $w_i[u,v]$. We adopted this design because the core role of OSIM is not to learn complex semantic representations, but rather to perform scoring and ranking of client updates at each parameter position. We will incorporate these formalized implementation details into the final version of our paper to ensure full reproducibility. Thank you again for your valuable feedback.
>
> ---
> >**Q8: PCA specificity: The proposed Poly-Consensus Aggregation seems generally applicable to any matrix aggregation, not specifically tailored to LoRA. Can author share more insights on how LoRA can be beneficial on PCA?**
>
> **A8:** We thank the reviewer for this important question, and we welcome the chance to clarify why PCA is indeed LoRA-specific.
>
> + **1. LoRA's Sensitivity Demands Precision:**
> Unlike dense, full-parameter updates, LoRA's low-rank updates are inherently sparse and highly sensitive to directional misalignment. Simple averaging often proves insufficient, as it can easily dilute or misdirect these updates under data heterogeneity. This sensitivity necessitates a more precise, learnable aggregation strategy. Our PCA mechanism provides this precision by adaptively assigning consensus weights to each parameter position, allowing it to correctly align and fuse the sensitive LoRA updates and effectively correct for client drift.
>
>
> + **2. Synergy with the LoRA-Specific Framework:**
> PCA is not designed in a vacuum but is integrated within our alternating freezing framework, which is a baseline strategy specifically for mitigating LoRA's inherent aggregation noise. PCA builds upon this foundation to address the more nuanced challenge of aggregation distortion that simple averaging.

---

> > ### Author Response · Authors · 2025-11-24
> > **Point-to-point response to 8fzm (6/6)**
> >
> > >**Q9: Assumption in Theorem 3.1: Is it realistic to assume the symmetric part of $R_{global}^T R_{client}$ is positive definite?**
> >
> > **A9:**  We thank the reviewer for this insightful technical question. We contend that this assumption is realistic as it is a natural consequence of our specific training dynamic, where $R_{client}$ acts as a residual correction to the frozen anchor $R_{global}$.
> >
> > + **1. Concise Explanation**
> > The assumption holds because $R_{client}$ is initialized to zero and trained to minimize local loss in the context of a frozen $R_{global}$. To effectively reduce loss, the optimizer inherently learns an $R_{client}$ that creates **constructive interference** with the global anchor. Learning an antagonistic $R_{client}$ (one that cancels out the global prior) would be a sub-optimal optimization path. Therefore, the assumption that the symmetric part of $R_{global}^T R_{client}$ is positive definite is simply the mathematical formalization of this expected **cooperative alignment**.
> >
> > + **2. Formal Justification**
> >   + **a. Constructive Alignment Condition:**
> > Since $R_{client}$ is learned as a residual to improve the representation provided by $R_{global}$, the transformation applied by $R_{client}$ should be synergistic with $R_{global}$ for a typical feature vector $x$. Geometrically, this implies that the update direction should align constructively with the anchor: $$\langle R_{global} x, R_{client} x \rangle > 0$$
> >   + **b. Matrix Formulation:**
> > Expanding the inner product yields:$$(R_{global} x)^T (R_{client} x) = x^T R_{global}^T R_{client} x > 0$$
> >   + **c. Connection to Positive Definiteness:**
> > The condition $x^T M x > 0$ for all non-zero $x$ is the definition of positive definiteness for the symmetric part of a matrix $M$. Thus, the cooperative nature of the residual learning implies: $$
> > \text{Sym}(R_{global}^T R_{client}) > 0 $$
> > This confirms that our assumption is a direct mathematical consequence of the optimizer seeking a constructive solution within our residual learning framework.
> >
> >
> >
> > We appreciate again your thoughtful review and we hope we addressed your concerns. Please let us know if you'd like any further information.

---

### Official Review · Reviewer_zbqX · 2025-11-03

**Soundness:** 4
**Presentation:** 4
**Contribution:** 3
**Rating:** 6
**Confidence:** 3

**Summary:**

In this paper, the authors propose RD-LoRA, a communication-efficient federated fine-tuning framework for large language models using parameter-efficient LoRA adapters, designed for realistic non-IID settings where clients have heterogeneous data distributions and even different LoRA ranks.  It introduces SCRD—a routing mechanism separating global and client-specific adaptation to prevent knowledge contamination—and PCA, an adaptive aggregation method that selects dominant client updates per parameter and stabilizes training with historical regularization.  The authors conduct experiments on Llama2-7B and TinyLlama show RD-LoRA outperforms existing methods on MMLU and MT-Bench.

**Strengths:**

1. The authors provide a practical problem formulation, and identify three concrete bottlenecks in scalable federated LoRA (aggregation noise, knowledge contamination, distortion) .

2.The authors introduce a routing matrix split into global (frozen) and client-specific (trainable) components, cleanly separating shared knowledge from local bias, with theoretical support for improved update stability.

3. The authors propose fine-grained aggregation via pca, which learns position-wise reliability weights and uses historical alignment to stabilize updates.

4. The paper measures and reports per round upload and download cost for each baseline.

**Weaknesses:**

1. Regarding experiments:

 a. the authors only adopt Llama2 7B and TinyLlama 1.1B as the backbone model, how about other models such as Qwen series model?

b. The training loop alternates B round and A round. But we only see two clients per round in experiments. How does that scale to a larger pool of clients across two hundred rounds.

c. Limited ablations on PCA internals:
   PCA has several moving parts. Poly Fusion Gate to create a contextual prior. Omni Scale Integration Module to learn per position weights. Historical steady alignment with a lambda term. The paper provides a single ablation that swaps PCA with FedAvg style averaging, and then discusses cosine similarity and singular value retention. That shows PCA helps, but it is still hard to tease apart which sub part of PCA is doing the heavy lifting. For example, is the historical regularizer alone already enough to stabilize training, or is the per position attention absolutely necessary. More granular ablations would make PCA easier to adopt by others.

**Questions:**

see weakness 1.a, 1.b, 1.c

---

> ### Author Response · Authors · 2025-11-24
> **Point-to-point response to zbqX (1/3)**
>
> We thank reviewer zbqX for the valuable time and constructive feedback. We provide point-to-point response below.
>
> >**Q1: The authors only adopt Llama2 7B and TinyLlama 1.1B as the backbone model, how about other models such as Qwen series model?**
>
> **A1:** We thank the reviewer for this valuable suggestion regarding the choice of base models. We fully acknowledge the rapid evolution of the open source LLM ecosystem and agree that validating our method on the latest architectures is crucial.
>
> + **1. Fair Comparison on Llama2-7B and TinyLlama-1.1B Models**
>
> We adopt Llama2-7B and TinyLlama-1.1B as base models, train them on Alpaca, Dolly and Wizard, and evaluate on MMLU and MT-Bench in order to ensure a fair and direct comparison with existing federated LoRA methods such as FlexLoRA[a] and FLoRA[b].
>
> + **2. New Experiments on Qwen3-8B and Llama4-17B**
>
> To address your concern and empirically demonstrate the robustness of our method, we conducted additional experiments using Qwen3-8B and Llama4-Scout-17B as base models. We maintained the same heterogeneous federated instruction tuning setting. The results, summarized in the tables below, demonstrate that RD-LoRA maintains a significant performance advantage even on these stronger baselines.
>
>    **Table B1.1: Performance Comparison on Qwen3-8B-Base (Heterogeneous Setting)**
>
>    | Method (Heter) |  Dolly (MMLU) | BigMath (AIME 25) |
>    | :--- |  :---: | :---: |
>    | **Zero-shot Base** |  72.49 | 17.11 |
>    | Zero-padding (FedIT) | 71.09 |   16.07   |
>    | FlexLoRA | 73.14 |    18.46   |
>    | **RD-LoRA (Ours)** | **74.11** | **19.73** |
>
>    **Table B1.2: Performance Comparison on Llama4-Scout-17B (Heterogeneous Setting)**
>
>    | Method (Heter) | Dolly (MMLU) | BigMath (AIME 25) |
>    | :--- | :---: | :---: |
>    | **Zero-shot Base** | 75.13 | 9.02 |
>    | Zero-padding (FedIT) | 73.64 |   8.85   |
>    | FlexLoRA | 75.67 |  11.03   |
>    | **RD-LoRA (Ours)** | **77.10** | **12.21** |
>
>   Notably, while baselines like Zero-padding (FedIT) suffer from performance degradation in heterogeneous settings (dropping below Zero-shot levels), our method consistently outperforms the strongest baseline, FlexLoRA. Specifically, RD-LoRA achieves gains of 0.97% and 1.27% on Qwen3-8B, and 1.43% and 1.18% on Llama4-17B for the Dolly and BigMath tasks, respectively.
>
> **References**
>
> - [a] Bai J, Chen D, Qian B, et al. Federated fine-tuning of large language models under heterogeneous tasks and client resources[J]. Advances in Neural Information Processing Systems, 2024, 37: 14457-14483.
> - [b] Wang Z, Shen Z, He Y, et al. Flora: Federated fine-tuning large language models with heterogeneous low-rank adaptations[J]. Advances in Neural Information Processing Systems, 2024, 37: 22513-22533.

---

> > ### Author Response · Authors · 2025-11-24
> > **Point-to-point response to zbqX (2/3)**
> >
> > >**Q2: The training loop alternates B round and A round. But we only see two clients per round in experiments. How does that scale to a larger pool of clients across two hundred rounds.**
> >
> > **A2:** We thank the reviewer for this important question on scalability. To address this, we conducted new experiments with expanded client pools. We evaluated two scenarios: a pool of 10 total clients (sampling 2, 4, and 8 per round) and a larger pool of 50 total clients (sampling 20 and 40 per round). The results, presented in the table below.
> >
> >   **Table B2.1: MMLU Performance on Dolly with an Expanded Client Pool (Llama2, Heterogeneous)**
> >
> > | Method             | Sampled 2 (Total 10) | Sampled 4 (Total 10) | Sampled 8 (Total 10) | Sampled 20 (Total 50) | Sampled 40 (Total 50) |
> > | :----------------- | :------------------: | :------------------: | :------------------: | :-------------------: | :-------------------: |
> > | FlexLoRA           |        45.91         |        47.46         |        48.83         |         46.11         |         47.57         |
> > | **RD-LoRA (ours)** |   **47.16 (+1.25)**  |   **49.14 (+1.68)**  |   **50.21 (+1.38)**  |   **47.87 (+1.76)**   |   **49.30 (+1.73)**   |
> >
> >
> >  The analysis confirms RD-LoRA's consistent superiority over SOTA FlexLoRA. Our method achieves both the peak accuracy of 50.21 (in the 10-client/8-participant setting) and the largest performance gap of 1.76% (in the 50-client/20-participant setting). This demonstrates that our performance advantage is maintained or even increased as more clients are sampled per round.

---

> > > ### Author Response · Authors · 2025-11-24
> > > **Point-to-point response to zbqX (3/3)**
> > >
> > > >**Q3: Limited ablations on PCA internals: PCA has several moving parts. Poly Fusion Gate to create a contextual prior. Omni Scale Integration Module to learn per position weights. Historical steady alignment with a lambda term. The paper provides a single ablation that swaps PCA with FedAvg style averaging, and then discusses cosine similarity and singular value retention. That shows PCA helps, but it is still hard to tease apart which sub part of PCA is doing the heavy lifting. For example, is the historical regularizer alone already enough to stabilize training, or is the per position attention absolutely necessary. More granular ablations would make PCA easier to adopt by others.**
> > >
> > > **A3:** We thank the reviewer for this insightful and constructive suggestion. We fully agree that a more fine-grained ablation of the PCA module is important for understanding its behaviour. Poly-Consensus Aggregation (PCA) mainly consists of two components: the Omni-Scale Integration Module (OSIM) and the Historical steady alignment term (Hist. Align). In the revised manuscript, we therefore add detailed ablation experiments with the following variants:
> > > - **Full RD-LoRA:** Our complete proposed method.
> > > - **RD-LoRA w/o OSIM:** The model without the per-position attention mechanism, but retaining historical alignment.
> > > - **RD-LoRA w/o Hist. Align.:** The model without the historical steady alignment term ($\lambda$), but retaining OSIM.
> > > - **RD-LoRA w/o OSIM and Hist. Align:** The baseline from our original paper, where the entire PCA module is replaced with FedAvg. The results of tkhis fine-grained comparison are presented below:
> > >
> > >   **Table B3.1: Granular Ablation Study of PCA Components (Llama2, Heterogeneous)**
> > >
> > >     | Method | Wizard (MMLU) | Dolly (MMLU) | Alpaca (MMLU) |
> > >     | :--- | :---: | :---: | :---: |
> > >     | RD-LoRA w/o OSIM and Hist. Align | 36.23 | 46.26 | 36.35 |
> > >     | RD-LoRA w/o Hist. Align. | 36.41 | 46.74 | 36.66 |
> > >     | RD-LoRA w/o OSIM | 36.28 | 46.31 | 36.37 |
> > >     | **RD-LoRA (Ours)** | **36.54** | **47.16** | **37.21** |
> > >
> > >   Based on the granular ablation study, we draw the following conclusions:
> > >
> > > + **Both Components are Essential and Synergistic:** The results confirm that both OSIM and Historical Alignment contribute positively. Removing either component degrades performance compared to the full RD-LoRA model. For instance, on the Alpaca dataset, the full model achieves 37.21, while removing Historical Alignment drops the score to 36.66, and removing OSIM drops it further to 36.37.
> > >
> > > + **OSIM is the Primary Performance Driver:** The data clearly indicates that the Omni-Scale Integration Module (OSIM) is the most critical component. Removing OSIM consistently results in a larger performance drop than removing Historical Alignment. On Dolly, removing OSIM causes a 0.85% drop (from 47.16 to 46.31), whereas removing Historical Alignment only results in a 0.42% drop. OSIM performs fine-grained fusion by adaptively assigning position-wise weights, selecting the most reliable client contribution at each parameter location to mitigate aggregation distortion.
> > >
> > > + **Historical Alignment Acts as a Key Stabiliser:** The historical alignment term provides a crucial final performance boost. Adding it back to the model (i.e., moving from "RD-LoRA w/o Hist. Align." to the full pipeline) lifts the performance on Dolly from 36.66 to a peak of 37.21. Historical Steady Alignment stabilises global updates by regularising them against the previous global state, mitigating inter-round oscillation and preserving consistency.
> > >
> > > We appreciate again your thoughtful review and we hope we addressed your concerns. Please let us know if you'd like any further information.

---

> > > > ### Comment · Reviewer_zbqX · 2025-11-27
> > > >
> > > > The additional experiments about the number of clients and the ablation of PCA have addressed my concern. I will keep my positive score.

---

### Official Review · Reviewer_RtDT · 2025-11-09

**Soundness:** 3
**Presentation:** 3
**Contribution:** 2
**Rating:** 2
**Confidence:** 3

**Summary:**

This paper proposes a method named RD-LoRA, which aims to improve federated LLM fine-tuning using LoRA-based methods. RD-LoRA seeks to address the limitations and challenges of existing PEFT methods for federated tuning of LLMs, namely aggregation noise, knowledge contamination, and aggregation distortion. The authors claim that RD-LoRA can address all the aforementioned challenges simultaneously. The key aggregation method used by RD-LoRA is the Poly-Consensus Aggregation approach, which adaptively aligns global model weights with local model weights. Experimental results have been presented to demonstrate the effectiveness of RD-LoRA.

**Strengths:**

- The paper is well written and well motivated.
- Improving the performance and efficiency of federated PEFT over LLMs seems to be a reasonable research direction to pursue.
- The proposed RD-LoRA method is easy to follow and intuitive.
- The experimental results of the paper look promising.

**Weaknesses:**

- The authors proposed a relatively complex method, but the accuracy improvement appears to be marginal (if not very marginal) based on the main results shown in Table 1 and Table 2, especially compared to FlexLoRA.
- The experiments were conducted on somewhat outdated models, for example, Llama2. We already have Llama4 and even stronger open source models, so why still fine-tuning on Llama2?
- The fine-tuning tasks also seem a bit outdated; it would be more interesting to show results on more challenging benchmarks such as AIME 2025.
- It is not clear what the implementation and hyperparameter tuning overheads are when using RD-LoRA compared to other simpler methods.

**Questions:**

Please focus on addressing the concerns in Weaknesses.

- If the performance gain is marginal, why using RD-LoRA?
- What does the performance of RD-LoRA look like on say Qwen3 finetuning and for harder tasks, e.g., some reasoning benchmarks.
- What are the overheads introduced by RD-LoRA ?

---

> ### Author Response · Authors · 2025-11-24
> **Point-to-point response to RtDT (1/4)**
>
> We thank reviewer RtDT for the valuable time and constructive feedback. We provide point-to-point response below.
>
> >**Q1: The authors proposed a relatively complex method, but the accuracy improvement appears to be marginal (if not very marginal) based on the main results shown in Table 1 and Table 2, especially compared to FlexLoRA. / If the performance gain is marginal, why using RD-LoRA.**
>
> **A1:** We thank the reviewer for this insightful observation. The trade-off between methodological complexity and performance gains, especially when compared to SOTA FlexLoRA, is indeed a crucial consideration. We will now address this concern by detailing our method's advantages from the following perspectives.
>
> + **1. Greater Performance Gains on Dolly and Alpaca**
>
> Our method demonstrates substantial improvements across the majority of datasets. Notably, compared to the strongest baseline, FLexLoRA, it achieves an increase of 1.98% on Dolly and 1.75% on the Alpaca dataset.
>
>   **Table A1.1: Comparison of Computational Complexity and Performance on MMLU Benchmark**
>
>    | Method |  Dolly (TinyLlama)  | Alpaca (TinyLlama)  | Dolly (Llama2)  | Alpaca (Llama2)  |
>    | :--- | :--- | :--- | :--- | :--- |
>    | FlexLoRA | 60.77 | 47.22 | 45.72 | 35.83 |
>    | **RD-LoRA (ours)** |  **62.61 (+1.84)** | **48.71 (+1.49)** | **47.70 (+1.98)** | **37.58 (+1.75)** |
>
> + **2. Balancing Efficiency and Performance compared to FlexLoRA on Wizard**
>
>     **(1) Dataset characteristics:** Wizard focuses on multi-turn conversational instructions, whereas MMLU stresses factual, topic-centric questions; together they form a typical cross-task generalisation setting where performance is mainly driven by pre-training semantics and light alignment, and is relatively insensitive to FL aggregation details. Under this mismatch, all methods are near-saturated on MMLU, so their performance gaps are minimal.
>
>     **(2) Table Analysis:** As illustrated in the table below and **Figure 5** of the revision, our method outperforms FlexLoRA while incurring lower communication costs and computational complexity. This demonstrates a superior trade-off between system efficiency (communication and computation) and model performance. RD-LoRA outperforms FlexLoRA by up to 1.19% on MMLU while reducing server computation by **two orders of magnitude**, as it avoids the cubic ($\boldsymbol{d}^3$) complexity inherent to FlexLoRA.
>
>   **Table A1.2: Comparison of Computational Complexity and Performance**
>
>     | Method | Server Computation |Wizard (TinyLlama) | Wizard (Llama2)|
>     | :--- | :--- | :--- | :--- |
>     | FlexLoRA | $O(K d^2 r + K d^2 + \boldsymbol{d}^3)$| 51.07 | 36.07 |
>     | **RD-LoRA (Ours)** | **$O(Kdr+ Kr^2)$**|**52.26 (+1.19)** |  **36.68 (+0.61)** |
>     > **Note:** $K$ denotes the number of clients, $d$ represents the input / output dimension, and $r$ is the rank of LoRA, where $r ≪ d$, accuracy on the MMLU benchmark.
>
>
> + **3. Significant Improvement Over the Baseline**
>
> As shown in the table below, our method demonstrates substantial improvements over the Zero-padding baseline, achieving a maximum gain of 4.71%.
>
>   **Table A1.3: Complexity Comparison of Federated LoRA Fine-tuning Methods on MMLU Benchmark**
>
>    | Method | Wizard (TinyLlama)  | Dolly (TinyLlama)  | Alpaca (TinyLlama)  | Wizard (Llama2)  | Dolly (Llama2)  | Alpaca (Llama2) |
>    | :--- | :--- | :--- | :--- | :--- | :--- | :--- |
>    | Zero-padding (FedIT) | 49.33 | 58.80 | 44.67 | 34.73 | 44.11 | 32.87 |
>    | **RD-LoRA (Ours)** | **52.26 (+2.93)** | **62.61 (+3.81)** | **48.71 (+4.04)** | **36.68 (+1.95)** | **47.70 (+3.59)** | **37.58 (+4.71)** |
>
> + **4. Improved Performance on Larger Models**
>
> To verify the scalability of our approach, we extended our experiments to larger models, specifically Qwen3-8B and Llama4-17B. As shown in Tables 1 and 2, our method consistently delivers superior performance compared to the SOTA FlexLoRA, with gains reaching up to 1.43%.
>
>   **Table A1.4: Performance Comparison on Qwen3-8B-Base (Heterogeneous Setting)**
>    | Method (Heter) |  Dolly (MMLU) | BigMath (AIME 25) |
>    | :--- | :---: | :---: |
>    | FlexLoRA | 73.14 |    18.46   |
>    | **RD-LoRA (Ours)** | **74.11 (+0.97)** | **19.73 (+1.27)** |
>
>    **Table A1.5: Performance Comparison on Llama4-Scout-17B (Heterogeneous Setting)**
>    | Method (Heter) | Dolly (MMLU) | BigMath (AIME 25) |
>    | :--- | :---: | :---: |
>    | FlexLoRA | 75.67 |  11.03   |
>    | **RD-LoRA (Ours)** | **77.10 (+1.43)** | **12.21 (+1.18)** |

---

> > ### Author Response · Authors · 2025-11-24
> > **Point-to-point response to RtDT (2/4)**
> >
> > >**Q2: The experiments were conducted on somewhat outdated models, for example, Llama2. We already have Llama4 and even stronger open source models, so why still fine-tuning on Llama2. / What does the performance of RD-LoRA look like on say Qwen3 finetuning and for harder tasks, e.g., some reasoning benchmarks.**
> >
> > **A2:** We thank the reviewer for this valuable suggestion regarding the choice of base models. We fully acknowledge the rapid evolution of the open source LLM ecosystem and agree that validating our method on the latest architectures is crucial. We have clarified the rationale for our initial selection of Llama2-7B and TinyLlama-1.1B. Furthermore, to demonstrate the generalisability of our approach, we have conducted additional experiments employing the more recent Qwen3-8B and Llama4-17B as our base models.
> >
> > + **1. Fair Comparison on Llama2-7B and TinyLlama-1.1B Models**
> >
> > We adopt Llama2-7B and TinyLlama-1.1B as base models, train them on Alpaca, Dolly and Wizard, and evaluate on MMLU and MT-Bench in order to ensure a fair and direct comparison with existing federated LoRA methods such as FlexLoRA [a] and FLoRA [b].
> >
> >
> > + **2. New Experiments on Qwen3-8B and Llama4-17B**
> >
> > To address your concern and empirically demonstrate the robustness of our method, we conducted additional experiments using Qwen3-8B and Llama4-Scout-17B as base models. We maintained the same heterogeneous federated instruction tuning setting. The results, summarized in the tables below, demonstrate that RD-LoRA maintains a significant performance advantage even on these stronger baselines. Our experiments compared the performance of four methods: the Zero-shot Base, Zero-padding (FedIT), FlexLoRA, and our proposed method, RD-LoRA.
> >
> >   **Table A2.1: Performance Comparison on Qwen3-8B-Base (Heterogeneous Setting)**
> >    | Method (Heter) |  Dolly (MMLU) | BigMath (AIME 25) |
> >    | :--- |  :---: | :---: |
> >    | Zero-shot Base |  72.49 | 17.11 |
> >    | Zero-padding (FedIT) | 71.09 |   16.07   |
> >    | FlexLoRA | 73.14 |    18.46   |
> >    | **RD-LoRA (Ours)** | **74.11** | **19.73** |
> >
> >   **Table A2.2: Performance Comparison on Llama4-Scout-17B (Heterogeneous Setting)**
> >    | Method (Heter) | Dolly (MMLU) | BigMath (AIME 25) |
> >    | :--- | :---: | :---: |
> >    | Zero-shot Base | 75.13 | 9.02 |
> >    | Zero-padding (FedIT) | 73.64 |   8.85   |
> >    | FlexLoRA | 75.67 |  11.03   |
> >    | **RD-LoRA (Ours)** | **77.10** | **12.21** |
> >
> >   Notably, while baselines like Zero-padding (FedIT) suffer from performance degradation in heterogeneous settings (dropping below Zero-shot levels), our method consistently outperforms the SOTA FlexLoRA. Specifically, RD-LoRA delivers improvements of up to 1.27% on Qwen3-8B and 1.43% on Llama4-17B. Compared to the Zero-shot Base, RD-LoRA achieves significant performance uplifts, delivering a maximum absolute gain of +2.62% on Qwen3-8B and +3.19% on Llama4-17B. When benchmarked against the Zero-padding (FedIT) method, RD-LoRA shows a substantial advantage, surpassing it by a maximum of +3.66% on Qwen3-8B and +3.46% on Llama4-17B.
> >
> > **References**
> > - [a] Bai J, Chen D, Qian B, et al. Federated fine-tuning of large language models under heterogeneous tasks and client resources[J]. Advances in Neural Information Processing Systems, 2024, 37: 14457-14483.
> > - [b] Wang Z, Shen Z, He Y, et al. Flora: Federated fine-tuning large language models with heterogeneous low-rank adaptations[J]. Advances in Neural Information Processing Systems, 2024, 37: 22513-22533.

---

> > > ### Author Response · Authors · 2025-11-24
> > > **Point-to-point response to RtDT (3/4)**
> > >
> > > >**Q3: The fine-tuning tasks also seem a bit outdated; it would be more interesting to show results on more challenging benchmarks such as AIME 2025. / What does the performance of RD-LoRA look like on say Qwen3 finetuning and for harder tasks, e.g., some reasoning benchmarks.**
> > >
> > > **A3:** We thank the reviewer for their valuable suggestion. We agree that validating our method on the latest architectures is crucial, especially given the rapidly evolving landscape of open-source LLMs.
> > >
> > > + **1. Experimental Set-up for Fair Comparison**
> > >
> > > We adopt the Alpaca, Dolly and Wizard datasets and evaluate on MMLU and MT-Bench to enable a fair and direct comparison with existing state-of-the-art methods such as FlexLoRA [a] and FLoRA [b]. By aligning our experimental setup with these works, all methods share the same evaluation protocol, which ensures fairness and supports reproducibility.
> > >
> > >
> > >
> > >
> > > + **2. New Experiments on the AIME 2025 Benchmark**
> > >
> > > To address your concern, we further trained Qwen3-8B and Llama4-Scout-17B on the BigMath dataset and evaluated them on the AIME 2025 benchmark under the same heterogeneous federated instruction-tuning setting. The results, reported in the tables below, show the corresponding performance of RD-LoRA and the baselines.
> > >
> > >   **Table A3.1: Performance Comparison on Qwen3-8B-Base (Heterogeneous Setting)**
> > >    | Method (Heter) |   BigMath (AIME 25) |
> > >    | :--- |  :---: |
> > >    | Zero-shot Base |   17.11 |
> > >    | Zero-padding (FedIT) |    16.07   |
> > >    | FlexLoRA |     18.46   |
> > >    | **RD-LoRA (Ours)** |  **19.73** |
> > >
> > >   **Table A3.2: Performance Comparison on Llama4-Scout-17B (Heterogeneous Setting)**
> > >    | Method (Heter) |  BigMath (AIME 25) |
> > >    | :--- | :---: |
> > >    | Zero-shot Base |  9.02 |
> > >    | Zero-padding (FedIT) |    8.85   |
> > >    | FlexLoRA |  11.03   |
> > >    | **RD-LoRA (Ours)** |  **12.21** |
> > >
> > >
> > > - Focusing on the BigMath (AIME 25) task, which demands complex reasoning, we observe that the FedIT baseline suffers from negative transfer, dropping below Zero-shot performance on both models. In contrast, our method significantly enhances reasoning capabilities in heterogeneous settings, outperforming the SOTA FlexLoRA, by 1.27% on Qwen3-8B and 1.18% on Llama4-17B.
> > > - Our method demonstrates a marked advantage over the Zero-shot Base, boosting reasoning performance by 2.62% on Qwen3-8B and a remarkable 3.19% on Llama4-17B.
> > > - Against the Zero-padding (FedIT) method, our approach establishes a substantial lead in reasoning capabilities, achieving improvements of 3.66% on Qwen3-8B and 3.36% on Llama4-17B.
> > >
> > > **References**
> > >
> > > - [a] Bai J, Chen D, Qian B, et al. Federated fine-tuning of large language models under heterogeneous tasks and client - resources[J]. Advances in Neural Information Processing Systems, 2024, 37: 14457-14483.
> > > - [b] Wang Z, Shen Z, He Y, et al. Flora: Federated fine-tuning large language models with heterogeneous low-rank adaptations[J]. Advances in Neural Information Processing Systems, 2024, 37: 22513-22533.

---

> > > > ### Author Response · Authors · 2025-11-24
> > > > **Point-to-point response to RtDT (4/4)**
> > > >
> > > > >**Q4: It is not clear what the implementation and hyperparameter tuning overheads are when using RD-LoRA compared to other simpler methods. / What are the overheads introduced by RD-LoRA?**
> > > >
> > > > **A4:** We thank the reviewer for this insightful question regarding practical overheads. To confirm RD-LoRA's practicality, we analysed its primary overheads. The analysis of client/server computation and uplink/downlink communication, detailed in the table, demonstrates that our method does not introduce significant costs.
> > > >
> > > >    **Table A4.1: Complexity Comparison of Federated LoRA Fine-tuning Methods**
> > > >   | Method | Client Computation | Server Computation | Uplink (UL) Comm. | Downlink (DL) Comm. |
> > > >   | :--- | :---: | :---: | :---: | :---: |
> > > >   | FFA-LoRA | $O(rd^2)$ | $O(Kdr)$ | $dr$ | $dr$ |
> > > >   | Zero-Padding (FedIT) | $O(r d^2)$ | $O(Kd)$ | $2dr$ | $2dr$ |
> > > >   | FLoRA | $O(r d^2)$ | $O(K d^2 r)$ | $2dr$ | $d^2$ |
> > > >   | LoRA-$\mathrm{A}^2$ | $O(r d^2)$ | $O(Kdr)$ | $dr$ | $dr$ |
> > > >   | FlexLoRA | $O(r d^2)$ | $O(K d^2 r + K d^2 + \boldsymbol{d}^3)$ | $2dr$ | $2dr$ |
> > > >   | **RD-LoRA (Ours)** | **$O(r d^2)$** | **$O(Kdr+ Kr^2)$** | **$dr + r^2$** | **$dr + r^2$** |
> > > >    > **Note:** $K$ denotes the number of clients, $d$ represents the input / output dimension, and $r$ is the rank of LoRA, where $r ≪ d$, Comm. = Communication.
> > > >
> > > >  As detailed in the table, we analyse the computational and communication overheads, demonstrating that our method maintains high efficiency without imposing significant costs.
> > > >
> > > > - **Client Computation:** Our method maintains a complexity of $O(rd^2)$, which is consistent with all baseline methods. This ensures that no additional computational burden is placed on resource-constrained client devices.
> > > > - **Server Computation:** Our server-side complexity is $O(Kdr + Kr^2)$. Given that $r \ll d$, this is on par with highly efficient methods like FFA-LoRA and LoRA-$\mathrm{A}^2$. This presents a marked efficiency improvement over both FLoRA and, in particular, FlexLoRA, which incurs a substantial cost of $O(Kd^2r + Kd^2 + \boldsymbol{d}^3)$. By circumventing the need for SVD operations on the full dimension $d$, our method avoids the cubic ($\boldsymbol{d}^3$) complexity inherent to FlexLoRA, thereby reducing server computation by **two orders of magnitude** and ensuring scalability.
> > > >
> > > > - **Communication:** Both uplink and downlink costs are $dr + r^2$. Given that $r \ll d$, the $r^2$ term is negligible. Consequently, our communication overhead is comparable to the most efficient baselines (FFA-LoRA) and is approximately half the cost of FlexLoRA ($2dr$).
> > > >
> > > > We appreciate again your thoughtful review and we hope we addressed your concerns. Please let us know if you'd like any further information.

---

### Author Response · Authors · 2025-12-02
**Summary of Response (1/2)**

We sincerely thank all reviewers for their insightful comments and constructive suggestions. We are encouraged that they recognise the importance of our method. To facilitate the decision-making process for the Area Chair, we summarise the key feedback and our major updates below.

**1. Summary of Reviewer Feedback**

-   **Reviewer RtDT** acknowledged our work as *"well motivated"* and *"easy to follow and intuitive"*, noting that *"the experimental results of the paper look promising"*.
-   **Reviewer zbqX** highlighted that our work *"provide[s] a practical problem formulation, and identif[ies] three concrete bottlenecks in scalable federated LoRA"* whilst *"cleanly separating shared knowledge from local bias, with theoretical support for improved update stability"*. Furthermore, after reviewing our response, this reviewer stated: *"The additional experiments have addressed my concern. I will keep my positive score."*
-   **Reviewer 8fzm** recognised that our method *"demonstrates clear empirical effectiveness and achieves superior performance over several strong baselines"* and praised us for *"providing a convincing empirical comparison"*.
-   **Reviewer wjiw** praised our method for having *"clear motivation"* and noted that the *"empirical results are strong and consistent across Llama-2 and TinyLlama in both homogeneous and heterogeneous rank settings"*.

**2. Revisions to the Manuscript**
We have updated the paper as suggested by the reviewers. **Changes are highlighted in the updated PDF.**

-   **Experiments on Larger Models:** We have added fine-tuning experiments using **Qwen3-8B** and **Llama4-7B** as base models. (Reviewer RtDT, zbqX)
-   **More Challenging Benchmarks:** New experiments on the difficult **AIME 2025** benchmark confirm our method maintains superior performance over SOTA FlexLoRA and significantly outperforms baselines on harder tasks. (Reviewer RtDT)
-   **Computational Complexity Analysis:** Our new complexity analysis compares RD-LoRA against advanced baselines in terms of computation and communication. RD-LoRA achieves significant efficiency gains, reducing complexity by two orders of magnitude compared to FLoRA and FlexLoRA.(Reviewer RtDT)
-   **Client Scalability Experiments:** We expanded experiments to vary client participation rates (2, 4, and 8 out of 10; 20 and 40 out of 50). Results confirm that RD-LoRA consistently outperforms the SOTA FlexLoRA method across all settings. (Reviewer zbqX, 8fzm, wjiw)
-   **Fine-grained PCA Ablation:** We added an ablation study for PCA's internal components. Results confirm that both OSIM and Historical Alignment contribute positively to performance. (Reviewer zbqX)
-   **200-Round Accuracy:** We added full 200-round curves on the Dolly dataset as **Figure 6** (averaged over five runs with standard deviation). RD-LoRA demonstrates superior convergence. (Reviewer wjiw)
-   **Random Seed Experiments:** We re-ran core Llama2-7B experiments five times with different seeds to rigorously evaluate result stability. (Reviewer 8fzm, wjiw)
-   **Zero-Shot Baselines:** We added zero-shot accuracy for all base models (e.g., Llama2, Qwen3) on MMLU, MT-Bench, and AIME 2025. RD-LoRA shows significant improvement over base models on each benchmark. (Reviewer 8fzm, wjiw)
-   **Refinement of Formulas:** We have redefined and clarified Equations (2) and (3), providing a clear mathematical explanation of the sources of aggregation noise. (Reviewer 8fzm)
-   **Theorem 3.2 (Global Convergence):** We introduced Theorem 3.2 to formally bridge the local analysis in Theorem 3.1 with global guarantees of knowledge separation. (Reviewer 8fzm)
-   **Justification for $R_{global}^T R_{client}$:** We have added an explanation justifying the validity of the assumption regarding $R_{global}^T R_{client}$. (Reviewer 8fzm)
-   **Full Description of OSIM:** We have added the formal formulas and a complete description of the implementation details for OSIM to ensure reproducibility. (Reviewer 8fzm)

---

> ### Author Response · Authors · 2025-12-02
> **Summary of Response (2/2)**
>
> **3. Key Clarifications**
> In our responses, we provided detailed explanations and evidence to address specific concerns:
>
> -   **FedIT Baseline:** The FedIT (FedAvg+LoRA) baseline was already included and evaluated as 'Zero-Padding'. To improve clarity, we have renamed it to **Zero-Padding (FedIT)** in the revision. (Reviewer 8fzm)
> -   **PCA Specificity:** LoRA's directional updates are easily diluted by simple averaging under heterogeneity. Our PCA mechanism uses adaptive, position-wise weighting to align and fuse these updates, effectively correcting client drift. (Reviewer 8fzm)
> -   **Zero-Padding & Aggregation Noise:** Our analysis shows zero-padding introduces no new aggregation noise. It serves only as a prerequisite for handling heterogeneous ranks, maintaining the same noise structure as the homogeneous case. (Reviewer wjiw)
> -   **Client Selection:** We use a standard partial participation protocol, **randomly sampling** 2 of 10 clients per round. This ratio is consistent across all methods, with updates and aggregation performed exclusively on the active subset. (Reviewer wjiw)
>
> **Conclusion:** Reviewers acknowledged our method's motivation and effectiveness but requested additional experiments for substantiation. We hope these clarifications and revisions address the reviewers' concerns. Once agin, we thank the Area Chair and reviewers for all efforts.

---

### Meta-Review · Area_Chair_ZUzz · 2026-01-06

**Summary:**

This paper proposes RD-LoRA for federated LoRA fine-tuning under non-IID data and heterogeneous LoRA ranks, combining (i) a Server–Client Routing Deconstructor (SCRD) and (ii) a Poly-Consensus Aggregation (PCA) module. Reviewers generally found the motivation clear, but the decision-relevant concerns were: (1) the pipeline is arguably over-engineered and evaluated under a very heavy training schedule, raising questions about practical relevance and attribution of gains; (2) the accuracy improvements are reported as marginal relative to strong baselines such as FlexLoRA; and (3) parts of the theory/analysis and experimental protocol details were initially unclear.

**Reviewer Concerns:**

Concerns addressed by the rebuttal / revision:

+Model/baseline breadth & harder tasks: Added experiments on newer base models (e.g., Qwen3-8B / Llama4) and a more challenging benchmark (AIME 2025), plus a complexity analysis.

+ Client scalability / partial participation: Added experiments varying participation rates and clarified the partial participation protocol.

+Stability and reporting rigor: Added multi-seed reruns / variance reporting and full-round curves (as claimed in the response summary and tables).

+PCA internals ablation: Provided more granular ablations; notably, Reviewer zbqX explicitly stated these additions addressed their concern and they would keep their positive score.

Concerns still outstanding (most decision-relevant):

- Practicality / complexity remains a core issue: Even with added analyses, the method still hinges on multiple interacting components (alternating A/B rounds, routing split, server-side PCA) and is evaluated with a heavy schedule (e.g., many rounds and local epochs), making it hard to judge whether the added machinery is justified for realistic deployments

-Incremental gains vs. strong baselines: A key reviewer concern is that improvements appear marginal—especially relative to FlexLoRA—so the contribution may not clear the bar for acceptance in a competitive cycle.

-Theory-to-practice gap (especially under federation): Theoretical arguments (e.g., around SCRD effectiveness and assumptions in Theorem 3.1) and the justification for heterogeneous-rank handling via zero-padding/truncation remain only partially convincing, given the acknowledged limitations and the reliance on pragmatic alignment tricks.

**Reviewer Scores:**

Reviewer RtDT (Rating: 2): Likely 2 → 3. The added experiments on newer models and harder benchmarks help, but the core critique about marginal gains relative to strong baselines remains.

Reviewer zbqX (Rating: 6): Likely stays at 6. This reviewer explicitly said the new client-number experiments and PCA ablations addressed their concerns and they would keep their positive score.

Reviewer wjiw (Rating: 4): Likely 4 → 4 (or at most 5). Added curves/seed runs and participation clarifications help, but the practicality/complexity concern is fundamental and unlikely to disappear via discussion.

Reviewer 8fzm (Rating: 4): Likely 4 → 5. The rebuttal substantially addresses clarity issues (Eq. (2)/(3), baseline naming/coverage, OSIM details), but residual concerns about theoretical grounding in the federated setting plausibly keep the score near threshold

---

### Decision · Program_Chairs · 2026-01-26

Reject